



# 1 Exceptional retreat of Novaya Zemlya's marine-terminating
# 2 outlet glaciers between 2000 and 2013

J. Rachel Carr[1], Heather Bell[2], Rebecca Killick[3], Tom Holt[4]
[1]School of Geography, Politics and Sociology, Newcastle University, Newcastle-upon-Tyne, NE1 7RU, UK
[2]Department of Geography, Durham University, Durham, DH13TQ, UK
[3]Department of Mathematics & Statistics, Lancaster University, LA1 4YF
[4]Centre for Glaciology, Department of Geography and Earth Sciences, Aberystwyth University, SY23 4RQ, UK.
*Correspondence to*: Rachel Carr (rachel.carr@newcastle.ac.uk)
**Abstract**
Novaya Zemlya (NVZ) has experienced rapid ice loss and accelerated marine-terminating glacier retreat during
the past two decades. However, it is unknown whether this retreat is exceptional longer-term and/or whether it
has persisted since 2010. Investigating this is vital, as dynamic thinning may contribute substantially to ice loss
from NVZ, but is not currently included in sea level rise predictions. Here, we use remotely sensed data to assess
controls on NVZ glacier retreat between the 1973/6 and 2015. Glaciers that terminate into lakes or the ocean
receded 3.5 times faster than those that terminate on land. Between 2000 and 2013, retreat rates were significantly
higher on marine-terminating outlet glaciers than during the previous 27 years, and we observe widespread slow-
down in retreat, and even advance, between 2013 and 2015. There were some common patterns in the timing of
glacier retreat, but the magnitude varied between individual glaciers. Rapid retreat between 2000-2013
corresponds to a period of significantly warmer air temperatures and reduced sea ice concentrations, and to
changes in the NAO and AMO. We need to assess the impact of this accelerated retreat on dynamic ice losses
from NVZ, to accurately quantify its future sea level rise contribution.
**1. Introduction**
Glaciers and ice caps are the main cryospheric source of global sea level rise and contributed approximately −215
±26 Gt yr$^{-1}$ between 2003 and 2009 (Gardner et al., 2013). This ice loss is predicted to continue during the 21$^{st}$
Century (Meier et al., 2007; Radić et al., 2014) and changes are expected to be particularly marked in the Arctic,
where warming of up to 8 °C is forecast (IPCC, 2013). Outside of the Greenland Ice Sheet, the Russian High
Arctic (RHA) accounts for approximately 20% of Arctic glacier ice (Dowdeswell and Williams, 1997; Radić et
al., 2014) and is, therefore, a major ice reservoir. It comprises three main archipelagos: Novaya Zemlya (glacier
area = 21,200 km$^2$), Severnaya Zemlya (16,700 km$^2$) and Franz Josef Land (12,700 km$^2$) (Moholdt et al., 2012).
Between 2003 and 2009, these glaciated regions lost ice at a rate of between 9.1 Gt a$^{-1}$ (Moholdt et al., 2012) and
11 Gt a$^{-1}$ (Gardner et al., 2013), with over 80% of mass loss coming from Novaya Zemlya (NVZ) (Moholdt et al.,
2012). This much larger contribution from NVZ has been attributed to it experiencing longer melt seasons and
high snowmelt variability between 1995 and 2011 (Zhao et al., 2014). More recent estimates suggest that the mass
balance of the RHA was -6.9 ± 7.4 Gt between 2004 and 2012 (Matsuo and Heki, 2013) and that thinning rates



increased to $-0.40 \pm 0.09$ ma$^{-1}$ between 2012/13-2014, compared to the long-term average of $-0.23 \pm 0.04$ m a$^{-1}$
(1952 and 2014) (Melkonian et al., 2016). The RHA is, therefore, following the Arctic-wide pattern of negative
mass balance (Gardner et al., 2013) and glacier retreat that has been observed in Greenland (Enderlin et al., 2014;
McMillan et al., 2016), Svalbard (Moholdt et al., 2010a; Moholdt et al., 2010b; Nuth et al., 2010), and the
Canadian Arctic (Enderlin et al., 2014; McMillan et al., 2016). However, the RHA has been studied far less than
other Arctic regions, despite its large ice volumes. Furthermore, assessment of 21[st] century glacier volume loss
highlights the RHA as one of the largest sources of future ice loss and contribution to sea level rise, with an
estimated loss of 20 – 28 mm of sea level rise equivalent by 2100 (Radić et al., 2014).
Arctic ice loss occurs via two main mechanisms: a net increase in surface melting, relative to surface
accumulation, and accelerated discharge from marine-terminating outlet glaciers (e.g. Enderlin et al., 2014; van
den Broeke et al., 2009). These marine-terminating outlets allow ice caps to respond rapidly to climatic change,
both immediately through calving and frontal retreat (e.g. Blaszczyk et al., 2009; Carr et al., 2014; McNabb and
Hock, 2014; Moon and Joughin, 2008) and also through long-term draw down of inland ice, often referred to as
'dynamic thinning' (e.g. Price et al., 2011; Pritchard et al., 2009). During the 2000s, widespread marine-
terminating glacier retreat was observed across the Arctic (e.g. Blaszczyk et al., 2009; Howat et al., 2008; McNabb
and Hock, 2014; Moon and Joughin, 2008; Nuth et al., 2007) and substantial retreat occurred on Novaya Zemlya
between 2000 and 2010 (Carr et al., 2014): retreat rates increased markedly from around 2000 on the Barents Sea
coast and from 2003 on the Kara Sea (Carr et al., 2014). Between 1992-2010, retreat rates on NVZ were an order
of magnitude higher on marine-terminating glaciers (-52.1 m a$^{-1}$) than on those terminating on land (-4.8 m a$^{-1}$)
(Carr et al., 2014), which mirrors patterns observed on other Arctic ice masses (e.g. Dowdeswell et al., 2008;
Moon and Joughin, 2008; Pritchard et al., 2009; Sole et al., 2008) and was linked to changes in sea ice
concentrations (Carr et al., 2014). However, the pattern of frontal position changes on NVZ prior to 1992 is
uncertain and previous results indicate different trends, dependant on the study period: some studies suggest
glaciers were comparatively stable or retreating slowly between 1964 and 1993 (Zeeberg and Forman, 2001),
whilst others indicate large reductions in both the volume (Kotlyakov et al., 2010) and the length of the ice coast
(Sharov, 2005) from ~1950 to 2000. Consequently, it is difficult to contextualise the observed period of rapid
retreat from ~2000 until 2010 (Carr et al., 2014), and to determine if it was exceptional or part of an ongoing
trend. Furthermore, it is unclear whether glacier retreat has continued to accelerate after 2010, and hence further
increased its contribution to sea level rise, or whether it has persisted at a similar rate. This paper aims to address
these limitations, by extending the time series of glacier frontal position data on NVZ to include the period 1973/76
to 2015, which represents the limits of available satellite data.
Initially, surface elevation change data from NVZ suggested that there was no significant difference in thinning
rates between marine- and land-terminating outlet glacier catchments between 2003 and 2009 (Moholdt et al.,
2012). This contrasted markedly with results from Greenland (e.g. Price et al., 2011; Sole et al., 2008), but was
similar to the Canadian Arctic, where the vast majority of recent ice loss occurred via increased surface melting
(~92% of total ice loss), rather than accelerated glacier discharge (~8 %) (Gardner et al., 2011). This implied that
outlet glacier retreat was having a limited and/or delayed impact on inland ice, or that available data were not
adequately capturing surface elevation change in outlet glacier basins (Carr et al., 2014). More recent results
demonstrate that thinning rates on marine-terminating glaciers on the Barents Sea coast are much higher than on



their land-terminating neighbours, suggesting that glacier retreat and calving does promote inland, dynamic
thinning (Melkonian et al., 2016). However, higher melt rates also contributed to surface lowering, evidenced by
the concurrent increase in thinning observed on land-terminating outlets (Melkonian et al., 2016). High rates of
dynamic thinning have also been identified on Severnaya Zemlya, following the collapse of the Matusevich Ice
Shelf in 2012 (Willis et al., 2015). Here, thinning rates increased to 3-4 times above the long-term average (1984-
2014), following the ice-shelf collapse in summer 2012, and outlet glaciers feeding into the ice shelf accelerated
by up to 200% (Willis et al., 2015). The most recent evidence, therefore, suggests that NVZ and other Russian
High Arctic ice masses are vulnerable to dynamic thinning, following glacier retreat and/or ice-shelf collapse.
Consequently, it is important to understand the longer-term retreat history on NVZ, in order to evaluate its impact
on future dynamic thinning. Furthermore, we need to assess whether the high glacier retreat rates observed on
NVZ during the 2000s have continued and/or increased, as this may lead to much larger losses in the future, and
may indicate that a step-change in glacier behaviour occurred in ~2000.
In this paper we use remotely sensed data to assess glacier frontal position change for all major (>1 km wide)
Novaya Zemlya outlet glaciers (Fig. 1). This includes all outlets from the northern ice cap and its subsidiary ice
caps (Fig. 1). We were unable to find the names of these subsidiary ice masses in the literature, so we name them
Sub 1 and Sub 2 (Fig. 1). A total of 54 outlet glaciers were investigated, which allowed us to assess the impact of
different glaciological, climatic and oceanic settings on retreat. Specifically, we assessed the impact of coast
(Barents versus Kara Sea on the northern ice mass), ice mass (northern ice cap, Sub 1 or Sub 2), terminus type
(marine-, lake- and land-terminating) and latitude (Table 1). The two coasts of Novaya Zemlya are characterised
by very different climatic and oceanic conditions: the Barents Sea coast is influenced by water from the north
Atlantic (Loeng, 1991; Pfirman et al., 1994; Politova et al., 2012) and subject to Atlantic cyclonic systems
(Zeeberg and Forman, 2001), which results in warmer air and ocean temperatures as well as higher precipitation
(Przybylak and Wyszyński, 2016; Zeeberg and Forman, 2001). In contrast, the Kara Sea coast is isolated from
north Atlantic weather systems, by the topographic barrier of NVZ (Pavlov and Pfirman, 1995), and is subject to
cold, Arctic-derived water, along with much higher sea ice concentrations (Zeeberg and Forman, 2001). We
therefore aim to investigate whether these differing climatic and oceanic conditions lead to major differences in
glacier retreat between the two coasts. Glaciers identified as surge-type (Grant et al., 2009) were excluded from
the retreat calculations and analysis. However, frontal position data are presented separately for three glaciers that
were actively surging during the study period. Glacier retreat was assessed from the 1973/6 to 2015, in order to
provide the greatest temporal coverage possible from satellite imagery. We use these data to address the following
questions:
1. At multi-decadal timescales, is there a significant difference in glacier retreat rates according to: i)

terminus type (land-, lake- or marine-terminating); ii) coast (Barents versus Kara Sea coast); iii) ice mass

(northern ice mass, Sub 1 or Sub 2) and; iv) latitude?

2. Are outlet glacier retreat rates observed between 2000 and 2010 on NVZ exceptional during the past ~

40 years?

3. Is glacier retreat accelerating, decelerating or persisting at the same rate?
4. Can we link observed retreat to changes in external forcing (air temperatures, sea ice and/or ocean

temperatures)?



## 2. Methods

### 2.1. Study area

This paper focuses on the ice masses located on the Severny Island, which is the northern island of the Novaya Zemlya archipelago (Fig. 1). The northern ice cap contains the vast majority of ice (19,841 km$^2$) and the majority of the main outlet glaciers (Fig. 1). The northern island also has two smaller ice caps, Sub 1 and Sub 2, which are much smaller in area (1010 km$^2$ and 705 km$^2$ respectively) and have far fewer, smaller outlet glaciers (Sub 1 = 4; Sub 2 = 5) (Fig. 1). We excluded all glaciers that have been previously identified as surge type and those smaller than 1 km in width from our analysis of glacier retreat rates. However, three glaciers were observed during their surge phase and are discussed separately. This resulted in a total of 54 outlet glaciers, which were located in a variety of settings and hence allowed us to assess spatial controls on glacier retreat (Table 1). The impact of coast could only be assessed for the main ice mass, as the glaciers on the smaller ice masses, Sub 1 and Sub 2, are located on the southern ice margin so do not fall on either coast (Fig. 1).

### 2.2. Glacier frontal position

Outlet glacier frontal positions were acquired predominantly from Landsat imagery. These data have a spatial resolution of 30 m and were obtained freely via the United States Geological Survey (USGS) Global Visualization Viewer (Glovis) (http://glovis.usgs.gov/). The frequency of available imagery varied considerably during the study period. Data were available annually from 1999 to 2015 and between 1985 and 1998, although georeferencing issues during the latter time period meant that imagery needed to be re-coregistered manually using stable, off-ice locations as tie-points. Prior to 1985, the only available Landsat scenes dated from 1973, and these also needed to be manually georeferenced. Hexagon KH-9 imagery was used to determine frontal positions in 1976 and 1977, but full coverage of the study area was not available for either year. The data resolution is 20 to 30 feet (~6-9 m). The earliest common date for which we have frontal positions for all glaciers is 1986, and so we calculate total retreat rates for the period 1986-2015 and use these values to assess spatial variability in glacier recession across the study region. All glacier frontal positions are calculated relative to 1986 (i.e. the frontal position in 1986 = 0 m), to allow for direct comparison.

Due to the lack of Landsat imagery during the 1990s, we use Synthetic Aperture Radar (SAR) Image Mode Precision data during this period. The data were provided by the European Space Agency and we use European Remote-sensing Satellite-1(ERS-1) and ERS-2 products (https://earth.esa.int/web/guest/data-access/browse-data-products/-/asset_publisher/y8Qb/content/sar-precision-image-product-1477). Following Carr et al. (2013b), the ERS scenes were first co-registered with ENVISAT imagery and then processed using the following steps: 1) apply precise orbital state vectors; radiometric calibration; multi-look; and terrain correction. This gave an output resolution of 37.5 m, which is comparable to Landsat. For each year and data type, imagery was acquired as close as possible to 31$^{st}$ July, to minimise the impact of seasonal variability. However, this is unlikely to substantially effect results, as previous studies suggest that seasonal variability in terminus position is very limited on NVZ (~100 m a$^{-1}$) (Carr et al., 2014) and is therefore much less than the interannual and inter-decadal variability we observe here. Glacier frontal position change was calculated using the box method: the terminus was repeatedly digitized from successive images, within a fixed reference box and the resultant change in area is divided by the reference box width, to get frontal position change (e.g. Moon and Joughin, 2008). Following previous studies



(Carr et al., 2014), we determined the frontal position errors for marine- and lake terminating outlets glaciers by
digitising 10 sections of rock coastline from six images, evenly spread through the time series (1976, 1986, 2000,
2005, 2010 and 2015) and across NVZ. The resultant error was 17.5 m, which equates to a retreat rate error of
1.75 m a$^{-1}$ at the decadal time intervals discussed here. The terminus is much harder to identify on land-terminating
outlet glaciers due to the similarity between the debris-covered ice margins and the surrounding land, which adds
an additional source of error. We quantified this by re-digitising a sub-sample of six land-terminating glaciers in
each of the six images, which were spread across NVZ. The additional error for land-terminating glaciers was
66.1 m, giving a total error of 68.4m, which equates to a retreat rate error of 6.86 m a$^{-1}$ for decadal intervals.
**2.3. Climate and ocean data**
Air temperature data were obtained from meteorological stations located on, and proximal to, Novaya Zemlya
(Fig. 1). Directly measured meteorological data are very sparse on NVZ and there are large gaps in the time series
for many stations. We use data from two stations, Malye Karmakuly and Im. E.K. Fedrova, as these are the closest
stations to the study glaciers that have a comprehensive (although still not complete) record during the study
period. The data were obtained from the Hydrometeorological Information, World Data Center Baseline
Climatological Data Sets (http://meteo.ru/english/climate/cl_data.php) and were provided at a monthly temporal
resolution. For each station, we calculated meteorological seasonal means (Dec-Feb, Mar-May, Jun-Aug, Sep-
Nov), in order to assess the timing of any changes in air temperature, as warming in certain seasons would have
a different impact on glacier retreat rates. Due to data gaps, particularly from 2013 onwards, we also assess
changes in air temperature using ERA-Interim reanalysis data (http://www.ecmwf.int/en/research/climate-
reanalysis/era-interim). We use temperature data from the surface (2 m elevation) and 850 m pressure level, as
these are likely to be a good proxy for meltwater availability (Fettweis, pers. Comm. 2017). We use the 'monthly
means of daily means' product, for all months between 1979 and 2015. As with the meteorological stations, we
calculate means for the meteorological seasons and annual means.
Sea ice data were acquired from the Nimbus-7 SMMR and DMSP SSM/I-SSMIS Passive Microwave dataset
(https://nsidc.org/data/docs/daac/nsidc0051_gsfc_seaice.gd.html). The data provide information on the
percentage of the ocean covered by sea ice and this is measured using brightness temperatures from microwave
sensors. The data have a spatial resolution of 25 x 25 km and we use the monthly-averaged product. This dataset
was selected due to its long temporal coverage, which extends from 26 October 1978 to 31 December 2015 and
thus provides a consistent dataset throughout our study period. Monthly sea ice concentrations were sampled from
the grid squares closest to the study glaciers and were split according to coast (i.e. Barents and Kara Sea). From
the monthly data, we calculated seasonal means and the number of ice free months, which we define as the number
of months where the mean monthly sea ice cover is less than 10%.
Data on the North Atlantic Oscillation (NAO) were obtained from The Climatic Research Unit
(https://crudata.uea.ac.uk/cru/data/nao/) and the monthly product was used. This records the normalized pressure
difference between Iceland and the Azores (Hurrell, 1995). Arctic Oscillation (AO) data were acquired from the
Climate Prediction Centre
(http://www.cpc.noaa.gov/products/precip/CWlink/daily_ao_index/teleconnections.shtml). The AO is
characterised by winds at 55°N, which circulate anticlockwise around the Arctic (e.g. Higgins et al., 2000; Zhou



et al., 2001). The AO index is calculated by projecting the AO loading pattern on to the daily anomaly 1000
millibar height field, at 20-90°N latitude (Zhou et al., 2001). The Atlantic Multidecadal Oscillation data (AMO)
is a mode of variability associated with averaged, de-trended SSTs in the North Atlantic and varies over timescales
of 60 to 80 years (Drinkwater et al., 2013; Sutton and Hodson, 2005). Monthly data were downloaded from the
National Oceanic and Atmospheric Administration (https://www.esrl.noaa.gov/psd/data/timeseries/AMO/).
We use ocean temperature data from the 'Climatological Atlas of the Nordic Seas and Northern North Atlantic'
(Hurrell, 1995; Korablev et al., 2014) (https://www.nodc.noaa.gov/OC5/nordic-seas/). The atlas compiles data
from over 500,000 oceanographic stations, located across the Nordic Seas, between 1900 and 2012. It provides
gridded climatologies of water temperature, salinity and density, at a range of depths (surface to 3500 m), for the
region bounded by 83.875 to 71.875 °N and 47.125°W to 57.875 °E. Here, we use data from the surface and 100m
depth, to capture changes in ocean temperatures at different depths: surface warming may influence glacier
behaviour through changes in sea ice and/or undercutting at the water-line (Benn et al., 2007), whereas warming
in the deeper layers can enhance sub-aqueous melting (Sutherland et al., 2013). A depth of 100 m was chosen, as
it is the deepest level that includes the majority of the continental shelf immediately offshore of Novaya Zemlya.
Further details of the data set production and error values are given in Korablev et al. (2014). We use the decadal
ocean temperature product to identify broad-scale changes, which is provided at the following time intervals:
1971-1980, 1981-1990, 1991-2000 and 2001-2012. We use the decadal product, as there are few observations
offshore of Novaya Zemlya during the 2000s, whereas the data coverage is much denser in the 1980s and 1990s
(a full inventory of the number and location of observations for each month and year is provided here:
https://www.nodc.noaa.gov/OC5/nordic-seas/atlas/inventory.html). As a result, maps of temperature changes in
the 2000s are produced using comparatively data few points, meaning that they may not be representative of
conditions in the region and that directly comparing data at a shorter temporal resolution (e.g. annual data) may
be inaccurate. Furthermore, the input data were measured offshore of Novaya Zemlya and not within the glacier
fjords. Consequently, there is uncertainty over the extent to which offshore warming is transmitted to the glacier
front and/or the degree of modification due to complexities in the circulation and water properties within glacial
fjords. We therefore use decadal-scale data to gain an overview of oceanic changes in the region, but we do not
attempt to use it for detailed analysis of the impact of ocean warming at the glacier front, nor for statistical testing.
**2.4. Statistical analysis**
We used a Kruksal Wallis test to investigate statistical differences in total retreat rate (1986-2015) for the different
categories of outlet glacier within our study population, i.e. terminus type (marine-, land- and lake-terminating),
coast (Barents and Kara Sea) and ice mass (northern ice cap, Sub 1 and Sub 2). The Kruksal Wallis test is a non-
parametric version of the one-way ANOVA (analysis of variance) test and analyses the variance using the ranks
of the data values, as opposed to the actual data. Consequently, it does not assume normality in the data, which is
required here, as Kolmogorov-Smirnov tests indicate that total retreat rate (1986-2015) is not normally distributed
for any of the glacier categories (e.g. terminus type). This is also the case when we test for normality at each of
the four time intervals discussed below (1973/6-1986, 1986-2000, 2000-2013 and 2013-2015). The Kruksal Wallis
test gives a p-value for the null hypothesis that two or more data samples come from the same population. As
such, a large p-value suggests it is likely the samples come from the same population, where as a small value



indicates that this is unlikely. We follow convention and use a significance value of 0.05, meaning that a p-value
of less than or equal to 0.05 indicates that the data samples are significantly different.
We assessed the influence of glacier latitude on total retreat rate (1986-2015), using simple linear regression. This
fits a line to the data points and gives an $R^2$ value and a p-value for this relationship. The $R^2$ value indicates how
well the line describes the data: if all points fell exactly on the line, the $R^2$ would equal 1, whereas if the points
were randomly distributed about the line, the $R^2$ would equal 0. The p-value tests the null hypothesis that the
regression coefficient is equal to zero, i.e. that the predictor variable (e.g. glacier catchment size) has no
relationship to the response variable (e.g. total glacier retreat rate). A p-value of 0.05 or less therefore indicates
that the null hypothesis can be rejected and that the predictor variable is related to the response variable (e.g.
glacier latitude is related to glacier retreat rate). The residuals for these regressions were normally distributed.
However, we also regressed catchment area against total retreat rate and the regression residuals were not normally
distributed, indicating that it is not appropriate to use regression in this case. Consequently, we used Spearman's
Rank Correlation Coefficient, which is non-parametric and therefore does not require the data to be normally
distributed. Catchments were obtained from (Moholdt et al., 2012).
Wilcoxon tests were used to assess significant differences in mean glacier retreat rates between four time intervals:
1973/6-1986, 1986-2000, 2000-2013 and 2013-2015. These intervals were chosen through manual assessment of
apparent breaks in the data. For each interval, data were split according to terminus type (marine, land and lake)
and marine-terminating glaciers were further sub-divided by coast (Barents and Kara Sea). For each category, we
then used the Wilcoxon test to determine whether mean retreat rates for all of the glaciers during one time period
(e.g. 1986-2000) were significantly different from those for another time period (e.g. 2000-2013). The Wilcoxon
test was selected as it is non-parametric and our retreat data are not normally distributed, and is suitable for testing
statistical difference between data from two time periods (Miles et al., 2013). As with the Kruksal Wallis test, a
p-value of less than or equal to 0.05 is taken as significant and indicates that the two time periods are significantly
different. We also used the Wilcoxon test to identify any significant differences in mean air temperatures and sea
ice conditions for the same time intervals as glacier retreat, to allow for direct comparison. For the first time
interval (1973/6-1986), we use air temperature data from 1976 to 1986 from the meteorological stations, but the
sea ice and ERA-Interim data are only available from 1979. The statistical analysis was done separately for sea
ice on the Barents and Kara Sea coast and using meteorological data from Malye Karmakuly and Im. E.K. Fedrova
(Fig. 1). ERA-Interim data was analysed as a whole, as the spatial resolution of the data does not allow us to
distinguish between the two coasts. In each case, we compared seasonal means for each year of a certain time
period, with the seasonal means for the other time period (e.g. 1976-1985 versus 2000-2012). For the sea ice data,
we used calendar seasons (Jan-Mar, Apr-Jun, Jul-Sep, Oct-Dec), which fits with the Arctic sea ice minima in
September and maxima in March. For the air temperature data, meteorological seasons (Dec-Feb, Mar-May, Jun-
Aug, Sep-Nov) are more appropriate. We also tested mean annual air temperatures and the number of sea-ice free
months.
In order to further investigate the temporal pattern of retreat on Novaya Zemlya, we use statistical changepoint
analysis (Eckley et al., 2011) We applied this to our frontal position data for marine- and lake-terminating glaciers,
and to the sea ice and air temperature data. Land-terminating glaciers are not included, due to the much higher
error margins compared to any trends, which could lead to erroneous changepoints being identified. Changepoint



analysis allows us to automatically identify significant changes in the time series data, and if there has been a shift
from one mode of behaviour to another (e.g. from slower to more rapid retreat) (Eckley et al., 2011). Formally, a
changepoint is a point in time where the statistical properties of prior data are different from the statistical
properties of subsequent data; the data between two changepoints is a segment. There are various ways that one
can determine when a changepoint should occur, but the most appropriate approach for our data is to consider
changes in regression.
In order to automate the process, we use the cpt.reg function in the R EnvCpt package (Killick et al., 2016) with
a minimum number of four data points between changes. This function uses the Pruned Exact Linear Time (PELT)
algorithm (Killick et al., 2012) from the changepoint package (Killick and Eckley, 2015) for fast and exact
detection of multiple changes. The function returns changepoint locations and estimates of the intercept and slope
of the regression lines between changes. We give the algorithm no information on when or how large a change
we might be expecting, allowing it to automatically determine statistically different parts of the data. In this way,
we use the analysis to determine if, and when, retreat rates change significantly on each of the marine- and lake-
terminating glaciers on NVZ, and whether there are any significant breaks in our sea ice and air temperature data.
We also apply the changepoint analysis to the number of sea ice free months, but as the data do not contain a
trend, we identify breaks using significant changes in the mean, rather than a change in regression. Thus, we can
identify any common behaviour between glaciers, the timing of any common changes, and compare this to any
significant changes in atmospheric temperatures and sea ice concentrations.
**3.    Results**
**3.1. Spatial controls on glacier retreat**
The Kruksal Wallis test was used to identify significant differences in total retreat rate (1986-2015) for glaciers
located in different settings. First, terminus type was investigated. Results demonstrated that total retreat rates
(1986-2015) were significantly higher on lake- and marine-terminating glaciers than those terminating on land, at
a very high confidence interval (<0.001) (Fig. 2). Retreat rates were 3.5 times higher on glaciers terminating in
water (lake = -49.1 m a$^{-1}$ and marine = -46.9 m a$^{-1}$) than those ending on land (-13.8 m a$^{-1}$) (Fig. 2). In contrast,
there was no significant difference between lake- and marine-terminating glaciers (Fig. 2). Next, we assessed the
role of coastal setting (i.e. Barents Sea versus Kara Sea) as climatic and oceanic conditions differ markedly
between the two coasts. When comparing glaciers with the same terminus type, there was no significant difference
in retreat rates between the two coasts (Fig. 2: p-value = 0.178 for marine-terminating glaciers and 1 for land-
terminating). Retreat rates on land-terminating glaciers were very similar on both coasts: Barents Sea = -6.5 m a$^{-1}$
and Kara Sea = -9.0 m a$^{-1}$ (Fig. 2). For marine-terminating outlets, retreat rates were higher on the Barents Sea
(-55.9 m a$^{-1}$) than on the Kara Sea (-37.2 m a$^{-1}$), but the difference was not significant (p=0.178) (Fig. 2). Results
confirmed that the significant difference in total retreat rates between land- and marine-terminating glaciers
persists when individual coasts are considered (Fig. 2). Finally, we tested for differences in retreat rate between
the ice caps of Novaya Zemlya, specifically the northern ice cap, which is by far the largest, and the two smaller,
subsidiary ice caps Sub 1 and Sub 2. Here, we found no significant difference in retreat rates between the ice
masses (Fig. 2). Retreat rates were highest on Sub 2, followed by the northern ice cap, and lowest on Sub 1 (Fig.
2). Our results therefore demonstrate that the only significant difference in total retreat rates (1986-2015) relates



to glacier terminus type, with land-terminating outlets retreating 3.5 times slower than those ending in lakes or
the ocean (Fig. 2).
We used simple linear regression to assess the relationship between total retreat rate (1986-2015) and latitude, as
there is a strong north-south gradient in climatic conditions on NVZ, but no significant linear relationship was
apparent ($R^2$ = 0.001 p = 0.819) (Fig. 3). However, if we divide the glaciers according to terminus type, total
retreat rate shows a significant positive relationship for land-terminating glaciers ($R^2$ = 0.363 p = 0.023), although
the $R^2$ value is comparatively small (Fig. 3). This indicates that more southerly land-terminating outlets are
retreating more rapidly than those in the north. Conversely, total retreat rate for lake-terminating glaciers has a
significant inverse relationship with total retreat rate ($R^2$ = 0.811 p = 0.014), suggesting that glaciers at high
latitudes retreat more rapidly (Fig. 3). No linear relationship is apparent between latitude and total retreat rate for
marine-terminating glaciers and the data show considerable scatter, particularly in the north (Fig. 3). We find no
significant relationship between catchment area and total retreat rate (RHO = -0.149 p = 0.339), which
demonstrates that observed retreat patterns are not simply a function of glacier size (i.e. that larger glacier retreat
more, simply because they are bigger).
**3.2. Temporal change**
Based on an initial assessment of the temporal pattern of retreat for individual glaciers, we manually identified
major break points in the data and divided glacier retreat rates into four time intervals: 1973/6 to 1986, 1986 to
2000, 2000 to 2013 and 2013 to 2015 (Fig. 4). Data were separated according to terminus type and, in the case of
marine-terminating glaciers, according to coast. We then used the Wilcoxon test to evaluate the statistical
difference between these time periods for each category (Table 2). For land- and lake-terminating glaciers, there
were no significant differences in retreat rates between any of the time periods (Fig. 4; Table 2). Indeed, retreat
rates on lake-terminating glaciers were remarkably consistent between 1986 and 2015, both over time and between
glaciers (Figs. 4 & 5). For marine-terminating glaciers on the Barents Sea coast, the periods 1973/6 – 1986 and
1986-2000 were not significantly different from each other and mean retreat rates were comparatively low (-20.5
and -22.3 m a$^{-1}$ respectively). In contrast, the periods 2000-2013 and 2013-2015 were both significantly different
to all other time intervals (Fig. 4; Table 2). Between 2000 and 2013, retreat rates were much higher than at any
other time (-85.4 m a$^{-1}$). Conversely, the average frontal position change between 2013 and 2015 was positive,
giving a mean advance of +11.6 m a$^{-1}$ (Fig. 4). On the Kara Sea coast, marine terminating outlet glacier retreat
rates were significantly higher between 2000 and 2013 than any other time period (-64.8 m a$^{-1}$) (Fig. 4; Table 2).
Retreat rates reduced substantially during the period 2013-2015 (-22.7 m a$^{-1}$) and were very similar to values in
1973/6-1986 (-27.2 m a$^{-1}$) and 1986-2000 (-22.4 m a$^{-1}$) (Fig. 4). On both the Barents and Kara sea coasts, the
temporal pattern of marine-terminating outlet glacier retreat showed large variability, both between individual
glaciers and over time (Fig. 5).
Following our initial analysis, we used changepoint analysis to further assess the temporal patterns of glacier
retreat, by identifying the timing of significant breaks in the data. On the Barents Sea coast, five glaciers underwent
a significant change in retreat rate from the early 1990s onwards (Fig. 6). Of these, retreat rates on four glaciers
(MAK, TAI2, VEL and VIZ; see Fig. 1 for glacier locations and names) subsequently increased, whereas retreat
was slower on INO between 1989 and 2006. The most widespread step-change on the Barents Sea coast occurred



in the early 2000s, after which nine glaciers retreated more rapidly (Fig. 6). A second widespread change in glacier
retreat rates occurred in the mid-2000s, which was also the second changepoint for four glaciers (Fig. 6). Of these
eight glaciers, only VOE retreated more slowly after the mid-2000s changepoint. On the Kara Sea coast, we see
a broadly similar temporal pattern, with two glaciers showing a significant change in retreat rate from the early
1990s, and again in 2005 and 2007 (Fig. 6). In the case of MG, retreat rates were higher after each breakpoint,
whereas for SHU1, retreat rates were lower between the 1990s and mid-2000s. Four glaciers began to retreat more
rapidly from 2000 onwards, and five other glaciers showed a significant change in retreat rates beginning between
2005 and 2010 (Fig. 6), with VER being the only glacier to show a reduction in retreat rates after this change (Fig.
6). Focusing on lake-terminating glaciers, a significant change in retreat rates began between 2006 and 2008 on
all but one glacier, which began to retreat more rapidly from 2004 onwards (Fig. 6).

### 353    3.3. Climatic controls

At Im. E.K. Fedrova, mean annual air temperatures were significantly warmer in 2000-2012 (-3.9 °C) than in
1976-1985 (-6.5 °C) or 1986-1999 (-6. 4°C) (Fig. 4; Table 3). Looking at seasonal patterns, air temperatures were
significantly higher during spring, summer and autumn in 2000-2012, compared to 1976-1985, and in summer,
autumn and winter, when compared with 1986-1999 (Fig, 4; Table 3). Summer air temperatures averaged 5.1 °C
in 2000-2012, compared to 3.8°C in 1986-1999 and 3.3°C in 1976-1985 (Fig. 4). Warming was particularly
marked in winter, increasing from -16.1°C (1976-1985) and -17.5°C (1986-1999) to -12.9°C in 2000-2012 (Fig.
4). Winter air temperatures then reduced to -15.9°C for the period 2013-2015 (Fig. 4), although this change was
not statistically significant (Table 3). A similar change in mean annual air temperatures was evident on Malye
Karmakuly, where temperatures were significantly higher in 2000-2012 (-3.1°C) than in 1976-1985 (-5.4°C) or
1986-1999 (-5.0°C) (Table 3; Fig 4). In all seasons, air temperatures were significantly higher in 2000-2012,
compared to 1976-1985 (Table 3), with the largest absolute increases occurring in winter (Fig. 4). However, only
autumn air temperatures were significantly warmer in 2000-2012 than 1986-1999 (Fig. 4; Table 3). No significant
differences in air temperatures were observed between 1976-1985 and 1986-1999 at either station (Table 3).
In the ERA-Interim reanalysis data, mean annual air temperatures increased significantly between 1986-1999 and
2000-2012 at both the surface and 850 m pressure level (Table 3). Winter (surface) and autumn (850 m)
temperatures also warmed significantly between these time intervals (Table 3). Surface air temperatures were
significantly warmer in 2013-2015, compared to 1986-1999, in winter and annually (Table 3). No significant
differences in air temperatures were observed at either height between 2000-2012 and 2013-2015 for any season
(Table 3). Surface air temperatures were comparable between 2000-2012 and 2013-2015 in winter and autumn,
and somewhat warmer in spring (+ 2.6°C) and summer (+0.7 °C) in 2013-2015 (Fig. 4). At 850m height, winter
(-0.7°C) and autumn temperatures were slightly cooler (-0.7°C) and summer temperatures were warmer (+0.8 °C)
in 2013-2015 than in 2000-2012 (Fig. 4). At the regional scale, warmer surface air temperatures penetrate further
into the Barents Sea and the southern Kara Sea with each time step (Supp. Fig. 1). We observed a similar, although
less marked, northward progression of the isotherms at 850 m height (Supp. Fig. 1).
On the Barents Sea coast, sea ice concentrations during all seasons were significantly lower in 2000-2012 than in
1976-1985 or 1986-1999, as was the number of ice free months (Fig. 7; Table 4). Between 1976-1985 and 2000-
2012, mean winter sea ice concentrations reduced from 68% to 35%, mean spring values declined from 59% to





28% and mean autumn averages fell from 27% to 7 % (Fig. 7). Mean summer sea ice concentrations reduced
slightly, from 12% to 5 % (Fig. 7). Over the same time interval, the number of ice free months increased from 3.0
to 6.9 (Fig. 7). Summer sea ice concentrations on the Barents Sea coast reduced significantly between 2000-2012
and 2013-2015, but no significant change was observed in any other month, nor in the number of ice free months
(Fig. 7; Table 4). With exception of winter, sea ice concentrations were significantly lower in 2013-2015 than in
1976-1985 or 1986-1999 (Fig.4; Table 4). As on the Barents Sea coast, sea ice concentrations on the Kara Sea
were significantly lower in all seasons in 2000-2012, compared to 1976-1985 or 1986-1999 (Fig. 7; Table 4).
Summer mean sea ice concentrations declined from 25% in 1976-1985, to 13% in 2000-2012 (Fig. 7). Over the
same time interval, autumn mean concentrations reduced from 56% to 33%, spring values declined from 87% to
73% and winter values decreased from 87% to 79% (Fig. 7). The number of ice free months also reduced from
1.6 (1976-1985) to 3.0 (2000-2012) (Fig. 7). No significant differences were apparent between seasonal sea ice
concentrations and the number of ice free months in 2013-2015 and any other time period, with the exception of
summer sea ice concentrations between 1976-1985 and 2013-2015 (Table 4).
Focusing on the changepoint analysis, we see a significant change in air temperatures at Im. E.K. Fedrova from
2008 onwards, after which air temperatures increased markedly (Fig. 6). On the Barents Sea coast, we observe
significant breaks in summer sea ice concentrations at 2000 and 2008: before 2000, summer sea ice showed a
downward trend, but large interannual variability; between 2000 and 2008, there was a slight upward trend and
much lower variability and; from 2008 onwards, summer sea ice concentrations were much lower, and showed
both a downward trend and limited interannual variability (Supp. Fig. 2). From 2005 onwards, we observed much
lower interannual variability in spring, summer and autumn sea ice concentrations (Supp. Fig. 2). After 2005,
summer sea ice concentrations on the Kara Sea coast showed much smaller interannual variability and had lower
values (Supp. Fig. 3). The number of ice free months increased significantly on both the Kara Sea (from 2003)
and Barents Sea (from 2005) (Fig. 6).
Between 1970 and 1989, the summer and annual NAO index were largely positive, with a few years of negative
values (Fig. 8A). From 1989 to 1994, values were all positive, followed by strongly negative values in 1995 (Fig.
8A). Subsequently, the summer and annual NAO index remained weakly negative between 1999 and 2012, with
values becoming increasingly negative in the final five years of this period (Fig. 8A). In 2013, the NAO index
became strongly positive, particularly during summer, and values were also positive in 2015 and 2016 (Fig. 8A).
The AO index follows an overall similar pattern to the NAO until ~2000, although shifts are less distinct: the
index is generally negative until 1988, followed by five years of more positive values. In the 2000s, the AO index
fluctuates between positive and negative, and more negative summer values are observed in 2009, 2011, 2014 and
2015 (Fig. 8B). The AMO was generally negative from 1970 – 2000, although values fluctuated and were positive
around 1990 (Fig. 8C). Subsequently, the AMO entered a positive phase from 2000 onwards (Fig. 8C).
At the broad spatial scale, data indicate that surface ocean temperatures have warmed in the Barents Sea over time
(Fig. 9). Warming was particularly marked in the area extending approximately 100 km offshore of the Barents
Sea coast and south of 76 °N. Here, temperatures ranged between 2 and 4 °C in 1971-1980 and reached up to 7
°C by 2001-2012 (Fig. 9), although it should be noted that data are much sparser for the latter period. The Kara
Sea also warmed over the study period, with temperatures increasing from 0-2 °C in 1971-1980, to 4-5 °C in
2001-2012 (Fig. 9). Although input data are comparatively sparse for 2001-2012, it appears that ocean



temperatures have warmed in both the Barents and Kara Seas at each time step, suggesting there may be a broad
scale warming trend in the region. At 100 m depth, the data suggest that warmer ocean water extends substantially
during the study period, on both the Barents and Kara Sea coasts (Fig.9).

### 3.4. Glacier surging

During the study period, we observed three glaciers surging: ANU, MAS and SER (Fig. 1). These were excluded
from the analysis of glacier retreat rates and are discussed separately here. ANU has previously been identified as
possibly surge-type, based on the presence of looped-moraine (Grant et al., 2009). Here, we identify an active
surge phase, on the basis of a number of characteristics identified from satellite imagery and following the
classification of Grant et al. (2009): rapid frontal advance, heavy crevassing and digitate terminus. High flow
speeds are also evident close to the terminus (Melkonian et al., 2016), which is consistent with the active phase
of surging. Our results show that advance began in 2008 and was ongoing in 2015, with the glacier advancing 683
m during this period (Fig. 10). MAS was previously confirmed as surge-type (Grant et al., 2009) and our data
suggest that its active phase persisted between 1989 and 2007 (Fig. 10A). The imagery indicates that surging on
MAS originates from the eastern limb of the glacier, which may be partially fed by the neighbouring glacier (Figs.
10B & C). This ice appears to have impacted on the eastern margin of the main outlet of MAS, causing glacier
advance and heavy crevassing on the eastern portion of its terminus (Figs. 10B & C). This explanation is consistent
with the lack of signs of surge type behaviour on the western margin of MAS (Figs. 10B & C) and considerable
visible displacement of ice and surface features on the eastern tributary (Figs. 10B & C). SRE was also confirmed
as a surge-type glacier by Grant et al. (2009), who suggested that glacier advance occurred between 1976/77 and
2001. Our results indicate that advance began somewhat later, sometime between July 1983 and July 1986, and
ended before August 2000 (Fig. 10A).

### 4. Discussion

### 4.1. Spatial patterns of glacier retreat

Our results demonstrate that retreat rates on marine terminating outlet glaciers ($-46.9$ m a$^{-1}$) were more than three
times higher than those on land ($-13.8$ m a$^{-1}$) between 1986 and 2015 (Fig. 2). This is consistent with previous,
shorter-term studies from Greenland (Moon and Joughin, 2008; Sole et al., 2008) and Svalbard (Dowdeswell et
al., 2008), which demonstrated an order of magnitude difference between marine- and land-terminating glaciers.
It also confirms that the differences in retreat rates, relating to terminus type, observed between 1992 and 2010
on NVZ (Carr et al., 2014) persist at multi-decadal timescales. Recent results suggest that marine-terminating
glacier retreat and/or ice tongue collapse can cause dynamic thinning in the RHA (Melkonian et al., 2016; Willis
et al., 2015), meaning that these long-term differences in retreat rates may lead to substantially higher thinning
rates in marine-terminating basins, at multi-decadal timescales. The Russian High Arctic is forecast to be the third
largest source of ice volume loss by 2100, outside of the ice sheets (Radić and Hock, 2011). However, these
estimates only account for surface mass balance, and not ice dynamics, meaning that they may underestimate 21st
Century ice loss for the RHA. Consequently, dynamic changes associated with marine-terminating outlet glacier
retreat on NVZ need to be taken into account, in order to accurately forecast its near-future ice loss and sea level
rise contribution.



Our data showed no significant difference in total retreat rates for marine-terminating (-46.9 m a$^{-1}$) and lake-
terminating glaciers (-49.1 m a$^{-1}$). This contrasts with results from Patagonia, which were obtained during a similar
time period (mid-1980s to 2001/11) and showed that lake-terminating outlet glaciers retreated significantly more
rapidly than those ending in the ocean (Sakakibara and Sugiyama, 2014). For example,  marine-terminating outlets
retreat at an average rate of -37.8 m a$^{-1}$ between 2000 and 2010/11, whereas lake-terminating glaciers receded at
-80.8 m a$^{-1}$ (Sakakibara and Sugiyama, 2014). Lake-terminating glacier retreat on NVZ also differs from
Patagonia, in that retreat rates are remarkably consistent between individual glaciers and remained similar over
time (Figs. 4 & 5). Conversely, frontal position changes in Patagonia showed major spatial variations and retreat
rates on several lake-terminating glaciers changed substantially between the two halves of the study period (mid-
1980's – 2000 and 2000-2010/11) (Sakakibara and Sugiyama, 2014).
One potential explanation for the common behaviour of the lake-terminating outlet glaciers on NVZ is that retreat
may be dynamically controlled and sustained by a series of feedbacks, once it has begun. As observed on large
Greenlandic tidewater glaciers, initial retreat may bring the terminus close to floatation, leading to faster flow and
thinning, which promote further increases in calving and retreat (e.g. Howat et al., 2007; Hughes, 1986; Joughin
et al., 2004; Meier and Post, 1987; Nick et al., 2009). This has been suggested as a potential mechanism for the
rapid recession for Upsala Glacier in Patagonia (Sakakibara and Sugiyama, 2014) and Yakutat Glacier, Alaska
(Trüssel et al., 2013). However, rapid retreat was not observed on all lake-terminating glaciers in Patagonia
(Sakakibara and Sugiyama, 2014) and the potential for these feedbacks to develop depends on basal topography
(e.g. Carr et al., 2015; Porter et al., 2014; Rignot et al., 2016). Consequently, the basal topography would need to
be similar for each of the NVZ glaciers to explain the very similar retreat patterns, which is not implausible, but
perhaps unlikely. Alternatively, it may be that the proglacial lakes act as a buffer for atmospheric warming, due
the greater thermal conductivity of water relative to air, and so reduce variability in retreat rates. Furthermore,
lake-terminating glaciers are not subject to variations in sea ice and ocean temperatures, which may account for
their more consistent retreat rates, compared to marine-terminating glaciers (Figs. 4 & 5). In order to differentiate
between these two explanations, data on lake temperature changes during the study period, and lake bathymetry
would be required. However, neither are currently available and we highlight this as an important area for further
research, given the rapid recession observed on these lake-terminating glaciers.
For the period between 1986 and 2015, we find no significant difference in retreat rates between the Barents and
Kara Sea coasts (Fig. 2). This is contrary to the results of a previous, shorter-term study, which showed that retreat
rates on the Barents Sea coast were significantly higher than on the Kara Sea between 1992 and 2010 (Carr et al.,
2014) and the higher thinning rates observed on marine outlets on the Barents Sea coast (Melkonian et al., 2016).
Furthermore, there are substantial differences in climatic and oceanic conditions on the two coasts (Figs. 4 & 7)
(Pfirman et al., 1994; Politova et al., 2012; Przybylak and Wyszyński, 2016; Zeeberg and Forman, 2001), so we
would expect to see significant differences in outlet glacier retreat rates. This indicates that longer-term glacier
retreat rates on NVZ may relate to much broader, regional scale climatic change, which is supported by the
widespread recession of glaciers across the Arctic during the past two decades (e.g. Blaszczyk et al., 2009; Carr
et al., 2014; Howat and Eddy, 2011; Jensen et al., 2016; Moon and Joughin, 2008). One potential overarching
control on NVZ frontal positions are fluctuations in the North Atlantic Oscillation (NAO), which covaries with
northern hemisphere air temperatures, Arctic sea ice and North Atlantic ocean temperatures (Hurrell, 1995;





Hurrell et al., 2003; IPCC, 2013). More recent work has also recognised the influence of the Atlantic Multidecadal
Oscillation (AMO) on oceanic and atmospheric conditions in the Barents Sea, and broader north Atlantic
(Drinkwater et al., 2013; Oziel et al., 2016). Our data suggest that the major phases of frontal position change on
NVZ correspond to changes the NAO and AMO (Fig. 8; Section 4.2.): rapid retreat between 2000-2013 coincides
with a weakly negative NAO and positive AMO, following almost three decades characterised by a generally
positive NAO and negative AMO (Fig. 8). As such, these large-scale changes may overwhelm smaller-scale
spatial variations between the two coasts of NVZ, when retreat is considered on multi-decadal time frames.
Marine-terminating outlet glacier retreat rates do not show a linear relationship latitude and there is considerable
scatter when the two variables are regressed (Fig. 3). This may be due to the influence of fjord geometry on glacier
response to climatic forcing (Carr et al., 2014) and the capacity for warmer ocean waters to access the calving
fronts. In contrast, southerly land-terminating outlets retreat more rapidly than those in the north, which we
attribute to the substantial latitudinal air temperature gradient on NVZ (Zeeberg and Forman, 2001). Conversely,
lake-terminating glaciers retreat more rapidly at more northerly latitudes (Fig. 3), which we speculate may relate
to the bathymetry and basal topography of individual glaciers, but data are not currently available to confirm this.
**4.2. Temporal patterns**
Our results show that retreat rates on marine-terminating outlet glaciers on NVZ were significantly higher between
2000 and 2013 than during the preceding 27 years (Fig. 4). At the same time, land-terminating outlets experienced
much lower retreat rates and did not change significantly during the study period (Figs. 4 & 5). This is consistent
with studies from elsewhere in the Arctic, which identified the 2000s as a period of elevated retreat on marine-
terminating glaciers (e.g. Blaszczyk et al., 2009; Howat and Eddy, 2011; Jensen et al., 2016; Moon and Joughin,
2008) and increasing ice loss (e.g. Gardner et al., 2013; Lenaerts et al., 2013; Moholdt et al., 2012; Nuth et al.,
2010; Shepherd et al., 2012). As discussed above, recent evidence suggests that glacier retreat in the Russian High
Arctic can trigger substantial dynamic thinning and ice acceleration (Melkonian et al., 2016; Willis et al., 2015),
but it not currently incorporated into predictions of 21[st] century ice loss from the region (Radić and Hock, 2011).
Consequently, the period of higher retreat rates during the 2000s may have a much longer-term impact on ice
losses from NVZ, and this needs to be quantified and incorporated into forecasts of ice loss and sea level rise
prediction.
Within the decadal patterns of glacier retreat, we observe clusters in the timing of significant changes in marine-
terminating glacier retreat rates (Fig. 6). Specifically, we see breaks in the frontal position time series on both the
Barents and Kara Sea coasts, beginning in the early 1990s, ~2000 and the mid-2000s (Fig. 6). This demonstrates
some synchronicity in changes in glacier behaviour around NVZ, although it is not ubiquitous (Fig. 6). The timing
of these changes coincides with those observed in Greenland, where the onset of widespread retreat and
acceleration in south-east Greenland began in ~2000 (e.g. Howat et al., 2008; Moon and Joughin, 2008; Seale et
al., 2011), and occurred from the mid-2000s onwards in the north-west (e.g. Carr et al., 2013b; Howat and Eddy,
2011; Jensen et al., 2016; McFadden et al., 2011; Moon et al., 2012). Whilst these changes could be coincidental,
they may also relate to broad, regional-scale changes observed in the North Atlantic region during the 2000s
(Beszczynska-Möller et al., 2012; Hanna et al., 2013; Hanna et al., 2012; Holliday et al., 2008; Sutherland et al.,
2013). Data demonstrate that the NAO was weakly negative from the mid-1990s until 2012, in contrast to strongly
positive conditions in the late 1980s and early 1990s, and the AMO was persistently positive from 2000 onwards,



following three decades of overall positive conditions (Fig. 8). These changes coincide with increases in glacier
retreat rates, sea ice decline and atmospheric warming in NVZ between 2000 and 2013 (Figs. 4 & 7).
Between the 1950s and mid-1990s, positive phases of the NAO were associated with the influx of warm Atlantic
Water into the Barents Sea (Hurrell, 1995; Loeng, 1991) and increased penetration of Atlantic cyclones and air
masses into the region, which lead to elevated air temperatures and precipitation (Zeeberg and Forman, 2001).
Conversely, negative NAO phases were associated with cooler oceanic and atmospheric conditions in the Barents
Sea (Zeeberg and Forman, 2001). During this period, therefore, the impact of the NAO was opposite in the Barents
Sea and in western portions of the Atlantic-influenced Arctic (e.g. the Labrador Sea) (Drinkwater et al., 2013;
Oziel et al., 2016). However, since the mid-1990s, changes in the Barents Sea and the western Atlantic Arctic
have been in phase, and warming and sea ice reductions have been widespread across both regions (Drinkwater
et al., 2013; Oziel et al., 2016). As such, increased glacier retreat rates on NVZ during the 2000s (Figs 4 & 5) may
have resulted from the switch to a weaker, and predominantly negative, NAO phase from the mid-1990s (Fig. 8),
which would promote warmer air and ocean temperatures, and reduced sea ice, as we observe in our data (Figs. 4
& 7). Previous studies have suggested a 3-5 year lag between NAO shifts and changes in conditions on NVZ, due
to the time required for Atlantic Water to transit into the Barents Sea (Belkin et al., 1998; Zeeberg and Forman,
2001), which is consistent with the onset of retreat in ~2000 (Figs. 4 & 8). However, it has recently been suggested
that the NAO's role may have reduced since the mid-1990s, and that the AMO may be the dominant influence on
warming in the North Atlantic (Drinkwater et al., 2013; Oziel et al., 2016). The AMO is thought to promote
blocking of high-pressure systems by westerly winds, which changes the wind field (Häkkinen et al., 2011). This
allows warm water to penetrate further into the Barents and other Nordic Seas, leading to atmospheric and oceanic
warming during periods with a weakly negative NAO (Häkkinen et al., 2011). As such, rapid retreat on NVZ
between 2000 and 2013 may have resulted from the combined effects of a weaker, more negative NAO from the
mid-1990s and a more positive AMO from 2000 onwards (Fig. 8). This suggests that synoptic climatic patterns
may be an important control on glacier retreat rates on NVZ and that the recent relationship between the NAO
and glacier change on NVZ contrasts with that observed during the 20th century (Zeeberg and Forman, 2001).
Following higher retreat rates in the 2000's, our data indicate that marine-terminating glacier retreat slowed from
2013 onwards on both the Barents and Kara Sea coasts, with several glaciers beginning to re-advance (Figs. 4 &
5). Our data demonstrate that marine-terminating glaciers on NVZ have previously undergone a step-like pattern
of retreat, with short (1-2 year) pauses in retreat (Fig. 5). Thus, it is unclear whether this reduction in retreat rates
is another temporary pause, before continued retreat, or the beginning of a new phase of reduced retreat rates. One
possible explanation for reduced retreat rates on both coasts of NVZ are the stronger NAO values observed from
the late 2000s onwards: winter 2009/10 had the most negative NAO for 200 years (Delworth et al., 2016; Osborn,
2011) and values were strongly positive in 2013 (Fig. 8A). This is consistent with the 3 to 5 year lag required for
NAO-related changes in Atlantic Water inflow to reach NVZ (Zeeberg and Forman, 2001) and so we speculate
that reduced glacier retreat rates from 2013 onwards (Figs. 4 & 5) may relate to an increase in the influence of the
NAO, relative to the AMO, from the late 2000s (Fig.8). Evidence indicates that the impact of the NAO in the
Barents Sea is now in-phase with the western North Atlantic (Drinkwater et al., 2013; Oziel et al., 2016), and so
a more positive NAO could lead to cooler conditions on NVZ, and hence glacier advance. However, the
relationship between large-scale features, such as the NAO and AMO, ocean conditions and glacier behaviour is





complex (Drinkwater et al., 2013; Oziel et al., 2016) and the period of glacier advance / reduced retreat on NVZ
has lasted only two years. Consequently, further monitoring is required to determine whether this represents a
longer-term trend, or a short-term change, and to confirm its relationship to synoptic climatic patterns.
Despite the changes in the NAO and AMO, our data show no significant change in sea ice concentrations, nor the
length of the ice free season, between 2000-2012 and 2013-2015 on either the Barents Sea or Kara Sea coast
(Table 4; Fig. 7). Likewise, we see no significant change in winter (Jan-Mar) air temperatures at Im. K. Fedorova
(Table 3; Fig. 4) nor in the ERA-Interim data during any season (Table3; Fig. 4). Although not significant, we see
summer warming of 0.7 °C (surface) and 0.8 °C (850 m pressure level) in the ERA-Interim data (Fig. 4), which
is the opposite of what we would expect if reductions air temperatures and surface melt were driving the slow-
down in retreat rates. As such, reduced retreat rates do not seem to be directly linked to short-term changes in sea
ice or air temperatures. They are also unlikely to result from changes in surface mass balance, as the response
time of NVZ glaciers is likely to be slow: they have long catchments (~40km), slow flow speeds (predominantly
<200 m a$^{-1}$ (Melkonian et al., 2016)) and are likely to be polythermal. Furthermore, thinning rates between 2012
and 2013/14 averaged 0.4 m a$^{-1}$ across the ice cap and reached up to 5 m a$^{-1}$ close to the glacier termini (Melkonian
et al., 2016), meaning that even a positive surface mass balance is very unlikely to deliver sufficient ice, quickly
enough, to promote advance and/or substantially lower retreat rates. Instead, this may be a response to oceanic
changes, which we cannot detect from available data, a lagged response and/or relate to more localised, glacier
specific factors. We suggest that the latter is unlikely, given the widespread and synchronous nature of the
observed reduction in retreat rates (Figs. 4 & 5). Future work should monitor retreat rates, to determine whether
reduced retreat is persistent, or is a short-term interruption to overall glacier retreat, and collect more extensive
oceanic data, to assess its impact on this change.
Although we observe some common behaviour, in terms of the approximate timing and general trend in retreat,
there is still substantial variability in the magnitude of retreat between individual marine-terminating glaciers
(Figs. 4 & 5). Furthermore, not all glaciers shared common changepoints and certain outlets showed a different
temporal pattern of retreat to the majority of the study population (Figs. 4-6). For example, INO retreated more
slowly between 1989 and 2006 than during the 1970s and 1980's. We attribute these differences to glacier-specific
factors, and, in particular, the fjord bathymetry and basal topography of individual glaciers. Previous studies have
highlighted the impact of fjord width on retreat rates on NVZ (Carr et al., 2014) and basal topography on marine-
terminating glacier behaviour elsewhere (e.g. Carr et al., 2015; Porter et al., 2014; Rignot et al., 2016). This may
result from the influence of fjord geometry on the stresses acting on the glacier, once it begins to retreat: as a fjord
widens, lateral resistive stresses will reduce and the ice must thin to conserve mass, making it more vulnerable to
calving (Echelmeyer et al., 1994; Raymond, 1996; van der Veen, 1998a & b), whilst retreat into progressively
deeper water can cause feedbacks to develop between thinning, floatation and retreat (e.g. Joughin and Alley,
2011; Joughin et al., 2008; Schoof, 2007). Thus, retreat into a deeper and/or wider fjord may promote higher
retreat rates on a given glacier, even with common climatic forcing. In addition, differences in fjord bathymetry
may determine whether warmer Atlantic Water can access the glacier front (Porter et al., 2014; Rignot et al.,
2016), which could promote further variations between glaciers. This highlights the need to collect basal
topographic data for NVZ outlet glaciers, which it is currently very limited, but a potentially key control on ice
loss rates.





### 4.3. Climatic and oceanic controls

Our data demonstrate that air temperatures were very substantially warmer between 2000 and 2012 than during the preceding decades, and that sea ice concentrations were also much lower on both the Barents and Kara Sea coasts during this period (Figs. 4 and 7). This is consistent with the atmospheric warming reported across the Arctic during the 2000s (e.g. Carr et al., 2013a; Hanna et al., 2013; Hanna et al., 2012; Mernild et al., 2013) and the well-documented decline in Arctic sea ice (Comiso et al., 2008; Kwok and Rothrock, 2009; Park et al., 2015). As such, the decadal patterns of marine-terminating outlet glacier retreat correspond to decadal-scale climatic change on NVZ (Figs. 4 & 7), and exceptional retreat during the 2000s coincided with significantly warmer air temperatures and lower sea ice concentrations (Tables 2 &3). Interestingly, step-changes in the air temperature and sea ice data identified by the changepoint analysis did not correspond to significant changes in outlet glacier retreat rates (Fig. 6), suggesting that such changes may not substantially influence retreat rates, or that the relationship may be more complex, e.g. due to lags in glacier response.

The much lower retreat rates on land-terminating outlets (Fig. 4) may indicate an oceanic driver for retreat rates on marine-terminating glaciers. Previous studies identified sea ice loss as a potentially important control on NVZ retreat rates (Carr et al., 2014), which fits with observed correspondence between sea ice loss and retreat, but it is unclear whether the two variables simply co-vary, or whether sea ice can drive ice loss, by extending the duration of seasonally high calving rates (e.g. Amundson et al., 2010; Miles et al., 2013; Moon et al., 2015). The available ocean data indicate that temperatures were substantially warmer during the 2000s (Fig. 9), which would provide a plausible mechanism for widespread retreat on both coasts of NVZ (Fig 4). However, oceanic data for the 2000s is sparse in the Barents and Kara Seas, compared to previous decades, so it is difficult to ascertain the magnitude and spatial distribution of warming, and to link it directly with glacier retreat patterns. Lake-terminating glaciers are not affected by changes in sea ice or ocean temperatures, but could be influenced by air temperatures. However, despite much higher air temperatures in the 2000s, mean retreat rates on lake-terminating outlet glaciers were similar for each decade of the study (Fig. 4), suggesting that the relationship is not straightforward. Instead, the presence of lakes may at least partly disconnect these glaciers from climatic forcing, by buffering the effects of air temperatures changes and/or by sustaining dynamic changes, following initial retreat (Sakakibara and Sugiyama, 2014; Trüssel et al., 2013).

### 4.4. Glacier Surging

During the study period, we identify three actively surging glaciers, based on various lines of glaciological and geomorphological evidence (Copland et al., 2003; Grant et al., 2009), including terminus advance (Fig. 10). Frontal advance persisted for 18 years on ANU and 15 years on SER, respectively, whilst ANU began to advance in 2008 and this continued until the end of the study period (Fig. 10). This is comparatively long for surge-type glaciers, which usually undergo short active phases over timeframes of months to years (Dowdeswell et al., 1991; Raymond, 1987). For comparison, surges on Tunabreen, Spitzbergen, last only ~2 years (Sevestre et al., 2015) and Basin 3 on Austfonna underwent major changes in its dynamic behaviour in just a few years (Dunse et al., 2015). Surges elsewhere can occur even more rapidly: the entire surge cycle of Variegated Glacier in Alaska takes approximately 1-2 decades and the active phase persists for only a few months (e.g. Bindschadler et al., 1977; Eisen et al., 2005; Kamb, 1987; Kamb et al., 1985; Raymond, 1987). Furthermore, the magnitude of advance on these three glaciers is in the order of a few hundred meters, which is smaller than advances associated with surges



on Tunabreen (1.4 km) and Kongsvegen (2 km) (Sevestre et al., 2015) and much less than the many kilometres of advance observed on Alaskan surge-type glaciers, such as Variegated Glacier (Bindschadler et al., 1977; Eisen et al., 2005). Consequently, the active phase on NVZ appears to be long, in comparison to other regions and terminus advance is more limited, which may provide insight into the mechanism(s) driving surging here and may indicate that these glaciers are located towards one end of the climatic envelope required for surging in the Arctic (Sevestre and Benn, 2015).

During the active phase of the NVZ surge glaciers, we observe large sediment plumes emanating from the glacier terminus (Fig. 9), which indicates that at least part of the glacier bed is warm-based during the surge. Together with the comparatively long surge interval, this supports the idea that changes in thermal regime may drive glacier surging on NVZ, as hypothesised for certain Svalbard glaciers (Dunse et al., 2015; Murray et al., 2003; Sevestre et al., 2015). In addition, the surge of MAS appears to have been triggered by a tributary glacier surging into it its lateral margin (Fig. 9). This demonstrates an alternative mechanism for surging, aside from changes in the thermal regime and/or hydrology conditions of the glacier, which has not been widely observed, but will depend strongly on the local glaciological and topographical setting of the glacier. The data presented here focus only on frontal advance and glaciological/geomorphological evidence, whereas information on ice velocities is also an important indicator of surging (Sevestre and Benn, 2015). Consequently, information on velocity and surface elevation changes are needed to further investigate the surge cycle and its possible controls on NVZ. This is important, as NVZ is thought to have conditions that are highly conducive to glacier surging (Sevestre and Benn, 2015), but has a long surge interval. We therefore want to ensure that we can disentangle surge behaviour and the impacts of climate change on NVZ.

### 5. Conclusions

At multi-decadal timescales, terminus type remains a major, over-arching determinant of outlet glacier retreat rates on NVZ. As observed elsewhere in the Arctic, land-terminating outlets retreated far more slowly than those ending in the ocean. However, we see no significant difference in retreat rates between ocean- and lake-terminating glaciers, which contrasts with findings in Patagonia. Retreat rates on lake-terminating glaciers were remarkably consistent between glaciers and over time, which may result from the buffering effect of lake temperature and/or the impact of lake bathymetry, which could facilitate rapid retreat that is largely independent of climate forcing, after an initial trigger. We cannot differentiate between these two scenarios with currently available data. Retreat rates on marine-terminating glaciers were exceptional between 2000 and 2013, compared to previous decades. However, retreat slowed on the vast majority of ocean-terminating glaciers from 2013 onwards, and several glaciers advanced, particularly on the Barents Sea coast. It is unclear whether this represents a temporary pause or a longer-term change, but it should be monitored in the future, given the potential for outlet glaciers to drive dynamic ice loss from NVZ. The onset of higher retreat rates coincides with a more negative, weaker phase of the NAO and a more positive AMO, whilst reduced retreat rates follow stronger NAO years. This suggests that synoptic atmospheric and oceanic patterns may influence NVZ glacier behaviour at decadal timescales. Marine-terminating glaciers showed some common patterns in terms of the onset of rapid retreat (1990s, ~2000 and mid 2000s), but showed substantial variation in the magnitude of retreat, which we attribute to glacier-specific factors. Glacier retreat corresponded with decadal-scale climate patterns: between 2000-2013, air temperatures were significantly warmer than the previous decades and sea ice concentrations were significantly



lower. Available data indicate oceanic warming, which could potentially explain why retreat rates on marine-terminating glaciers far exceed those ending on land, but data are comparatively sparse from 2000 onwards, making their relationship to glacier retreat rate difficult to evaluate. The surge phase on NVZ glaciers appears to be comparatively long, and warrants further investigation, to separate its impact on ice dynamics from that of climate-induced change and to determine the potential mechanism(s) driving these long surges. Recent results suggest that outlet glaciers can trigger dynamic losses on NVZ, but these processes are not yet included in estimates of the region's contribution to sea level rise. As such, it is vital to determine the longer-term impacts of exceptional glacier retreat during the 2000s and to monitor the near-future behaviour of these outlets.

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





| Characteristic | Category | Number of glaciers |
|---|---|---|
| **Coast** | Barents Sea | 29 |
| | Kara Sea | 19 |
| **Ice mass** | Northern ice mass | 43 |
| | Subsidiary ice mass 1 | 4 |
| | Subsidiary ice mass 2 | 5 |
| **Terminus type** | Marine | 32 |
| | Lake | 6 |
| | Land | 15 |

**Table 1**. Number of outlet glaciers contained within each category used to assess spatial variations in retreat rate, specifically coast, ice mass and terminus type.

| | Barents Sea marine-terminating | Kara Sea marine-terminating | Land-terminating | Lake-terminating |
|---|---|---|---|---|
| 76-86 / 86-00 | 0.440 | 0.538 | 0.982 | 0.486 |
| 76-86 / 00-13 | **>0.001** | **0.018** | 0.085 | 0.686 |
| 76-86 / 13-15 | **0.008** | 0.497 | 0.945 | 0.686 |
| 86-00 / 00-13 | **0.001** | **0.008** | 0.223 | 0.886 |
| 86-00 / 13-15 | **0.001** | 0.935 | 0.909 | 0.886 |
| 00-13 / 13-15 | **>0.001** | **0.009** | 0.597 | 0.686 |

**Table 2.** Wilcoxon test results, used to assess significant differences in retreat rates between each manually-identified time interval (1976-1986, 1986-2000, 2000-2013, 2013, 2015). Retreat rate data were tested separately for each terminus type, and marine-terminating glaciers were further sub-divided by coast. Following convention, p-values of <0.05 are considered significant and are highlighted in bold.

| Station | Time interval | Season | | | | |
|---|---|---|---|---|---|---|
| | | DJF | MAM | JJA | SON | Annual |
| Im. E.K. Fedorova | 13-15 / 86-99 | 0.432 | | | | |
| Im. E.K. Fedorova | 13-15 / 76-85 | 0.937 | | | | |
| Im. E.K. Fedorova | 00-12 / 13-15 | 0.287 | | | | |
| Im. E.K. Fedorova | 00-12 / 86-99 | **0.011** | 0.643 | **0.043** | **0.008** | **0.013** |
| Im. E.K. Fedorova | 00-12 / 76-85 | 0.186 | **0.035** | **0.045** | **0.003** | **0.003** |
| Im. E.K. Fedorova | 86-99 / 76-85 | 0.188 | 0.089 | 0.704 | 0.495 | 0.828 |
| | | | | | | |
| Malye Karmakuly | 13-15 / 86-99 | | | | | |
| Malye Karmakuly | 13-15 / 76-85 | | | | | |
| Malye Karmakuly | 00-12 / 13-15 | | - | - | - | - |





| Malye Karmakuly | 00-12 / 86-99 | 0.017 | 0.840 | 0.056 | **0.007** | **0.017** |
| Malye Karmakuly | 00-12 / 76-85 | **0.038** | **0.041** | **0.045** | **0.004** | **>0.001** |
| Malye Karmakuly | 86-99 / 76-85 | 0.623 | 0.086 | 0.5977 | 0.673 | 0.212 |
| | | | | | | |
| ERA-Interim (surface) | 13-15 / 86-99 | **0.032** | 0.156 | 0.197 | 0.156 | **0.006** |
| ERA-Interim (surface) | 13-15 / 76-85 | 0.714 | 0.083 | 0.517 | 0.833 | 0.117 |
| ERA-Interim (surface) | 00-12 / 13-15 | 0.900 | 0.189 | 0.364 | 0.593 | 0.239 |
| ERA-Interim (surface) | 00-12 / 86-99 | **0.006** | 0.942 | 0.981 | 0.062 | **0.044** |
| ERA-Interim (surface) | 00-12 / 76-85 | 0.765 | 0.579 | 0.526 | 0.874 | 0.267 |
| ERA-Interim (surface) | 86-99 / 76-85 | 0.127 | 0.233 | 0.970 | 0.192 | 0.794 |
| | | | | | | |
| ERA-Interim (850 m) | 13-15 / 86-99 | 0.591 | 0.509 | 0.432 | 0.500 | 0.206 |
| ERA-Interim (850 m) | 13-15 / 76-85 | 0.548 | 0.383 | 0.833 | 0.733 | 0.383 |
| ERA-Interim (850 m) | 00-12 / 13-15 | 0.521 | 0.611 | 0.782 | 0.511 | 0.900 |
| ERA-Interim (850 m) | 00-12 / 86-99 | 0.062 | 0.752 | 0.058 | **0.041** | **0.004** |
| ERA-Interim (850 m) | 00-12 / 76-85 | 0.831 | 0.303 | 0.939 | 0.751 | 0.132 |
| ERA-Interim (850 m) | 86-99 / 76-85 | 0.149 | 0.433 | 0.433 | 0.146 | 0.576 |
| | | | | | | |

**Table 3.** P-values for Wilcoxon tests for significant differences in mean seasonal and mean annual air temperatures, for the periods 1976-1985, 1986-1999, 2000-2013, and 2013-2015. Following convention, p-values of <0.05 are considered significant and are highlighted in bold.

| Coast | Time interval | Season | | | | |
| --- | --- | --- | --- | --- | --- | --- |
| | | JFM | AMJ | JAS | OND | Ice-free months |
| Barents | 13-15 / 86-99 | **0.003** | **0.012** | **0.003** | **0.003** | **0.003** |
| Barents | 13-15 / 76-85 | 0.067 | **0.017** | **0.017** | **0.017** | **0.017** |
| Barents | 00-12 / 13-15 | 0.704 | 0.296 | **0.039** | 0.057 | 0.086 |
| Barents | 00-12 / 86-99 | **0.002** | **0.009** | **0.019** | **>0.001** | **0.001** |
| Barents | 00-12 / 76-85 | **0.006** | **0.002** | **0.002** | **0.001** | **0.002** |
| Barents | 86-99 / 76-85 | 0.279 | 0.080 | 0.218 | **0.179** | 0.213 |
| | | | | | | |
| Kara | 13-15 / 86-99 | 0.677 | 0.677 | 0.244 | 0.591 | 0.088 |
| Kara | 13-15 / 76-85 | 1 | 0.667 | **0.017** | 0.267 | 0.067 |
| Kara | 00-12 / 13-15 | 0.082 | 0.057 | 0.921 | 0.082 | 0.561 |
| Kara | 00-12 / 86-99 | **>0.001** | **>0.001** | **>0.001** | **>0.001** | **0.037** |
| Kara | 00-12 / 76-85 | **>0.001** | **>0.001** | **>0.001** | **>0.001** | **0.011** |

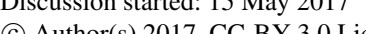



| Kara | 86-99 / 76-85 | **0.003** | **0.034** | **0.028** | **0.001** | 0.300 |
|------|---------------|-----------|-----------|-----------|-----------|-------|

**Table 4.** P-values for Wilcoxon tests for significant differences in mean seasonal sea ice concentrations and the
number of ice-free months, for the periods 1976-1985, 1986-1999 and 2000-2013. Following convention, p-
values of <0.05 are considered significant and are highlighted in bold.





**Figures**

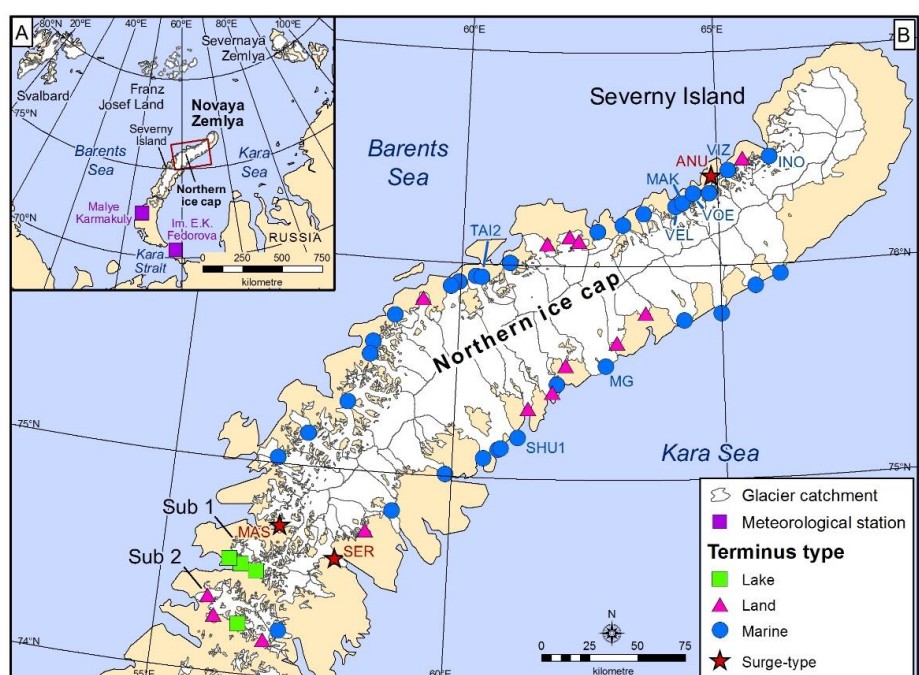


**Figure 1:** Location map, showing the study area and outlet glaciers. A) Location of Novaya Zemlya, in relation
to major land and water masses. Meteorological stations where air temperature data were acquired are indicated
by a purple square. B) Study glacier locations and main glacier catchments (provided by G. Moholdt and available
via GLIMS database). Glaciers are symbolised according to terminus type: marine terminating (blue circle); land-
terminating (pink triangle); lake terminating (green square); and observed surging during the study period (red
star). Glaciers observed to surge are: Anuchina (ANU), Mashigina (MAS), and Serp i Molot (SER).






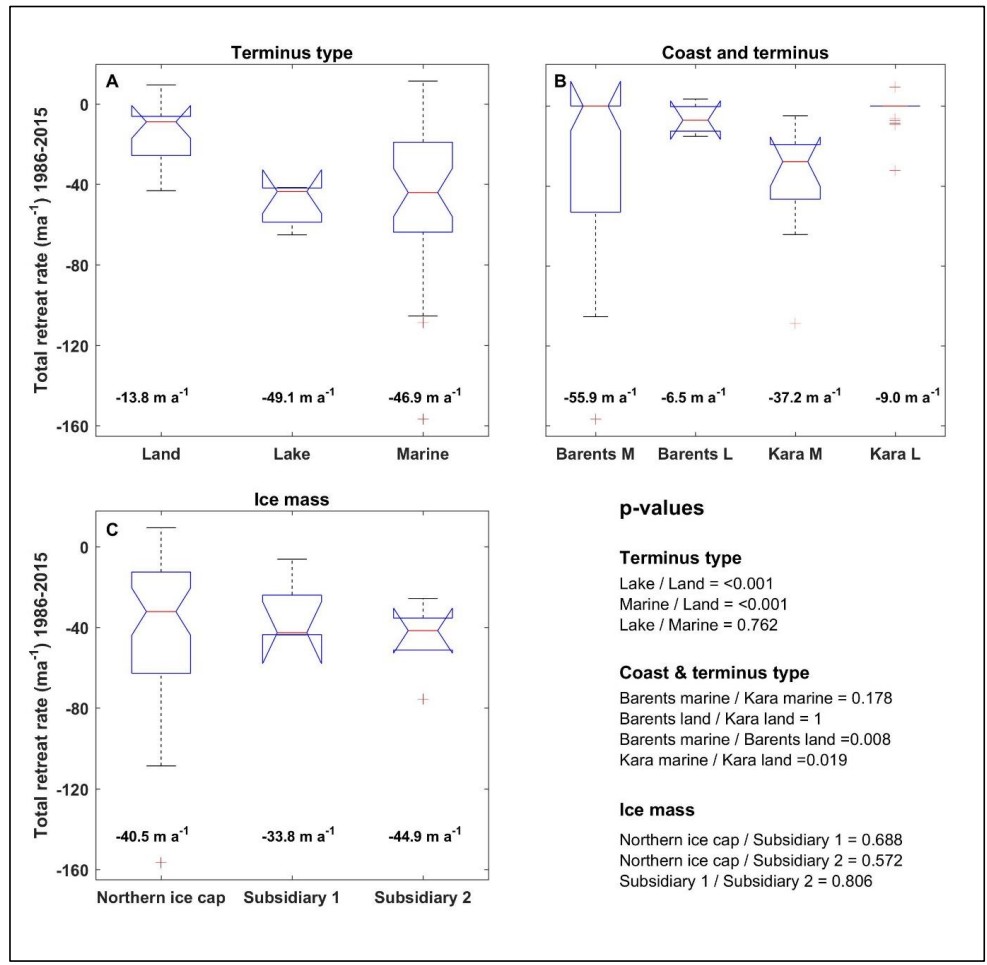


**Figure 2.** Box plots and Kruksal Wallis test results for different glacier terminus settings, for: A) terminus type; B) coast and terminus, L = land-terminating, m = marine-terminating; and C) ice mass, specifically the northern ice cap and subsidiary ice caps 1 and 2. See Figure 1 for ice cap locations. In all cases, total retreat rate (1986-2015) is used to test for significant differences between the classes. Mean total retreat rates for each class are given on each plot, below the associated box plot. For each box plot, the red central line represents the median, the blue lines the upper and lower quartile, red crosses are outliers (a value more than 1.5 times the interquartile range above / below the interquartile values) and the black lines are the whiskers, which extend from the interquartile ranges to the maximum values that are not classed as outliers. P-values for each Kruksal Wallis test are given on the right of the plot.

982

983



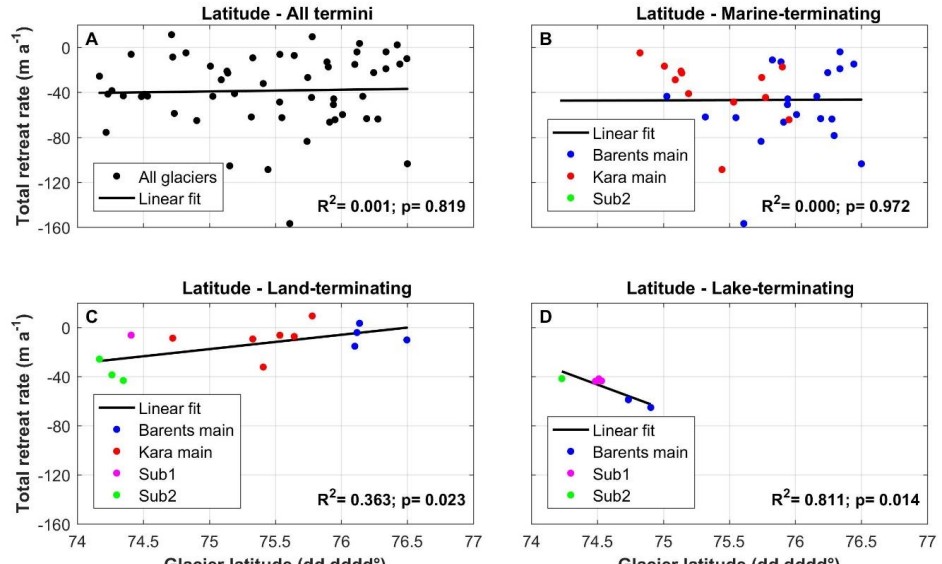

**Figure 3.** Linear regression of total retreat rate (1986-2015) versus glacier latitude. Latitude was regressed against total glacier retreat rate for A) All outlet glaciers in the study sample; B) marine-terminating glaciers only; C) land-terminating glaciers only; D) lake-terminating glaciers only. In all cases, the linear regression line is shown, as are the associated $R^2$ and p-values. The $R^2$ value indicates how well the line describes the data and the p-value indicates the significance of the regression coefficients, i.e. the likelihood that the predictor and response variable are unrelated.

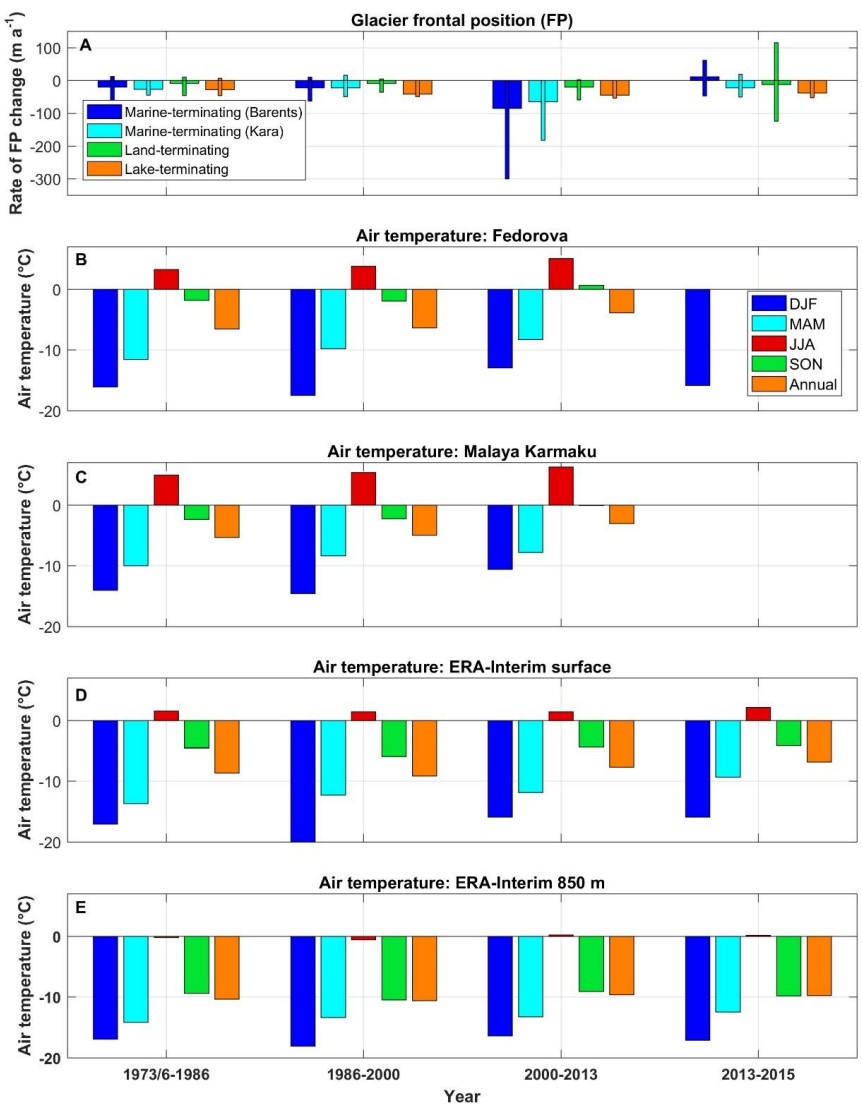

**Figure 4.** Mean retreat rates for Novaya Zemlya outlet glaciers, and mean air temperatures at Im. K. Fedorova
and Malaya Karmaku (Fig. 1). Data are split into four time periods, based on manually identified breaks in the
glacier retreat data: 1973/6-1986, 1986-2000, 2000-2013 and 2013-2015. A) Retreat rates were calculated
separately for different terminus types and marine-terminating glaciers were further sub-divided into those
terminating into the Barents Sea versus the Kara Sea. Wide bars represent mean values and thin bars represent the
total range (i.e. minimum and maximum values) within each category. B-E) Mean seasonal air temperatures (Dec-
Feb, Mar-May, Jun-Aug and Sep-Nov) and mean annual air temperatures for Im. K. Fedrova (B), Malaya
Karmaku (C), ERA-Interim surface (D) and ERA-Interim 850 m pressure level (E). Note that only mean values
for Im. K. Fedorova in Jan-Mar are calculated for 2013-2015, due to data availability.





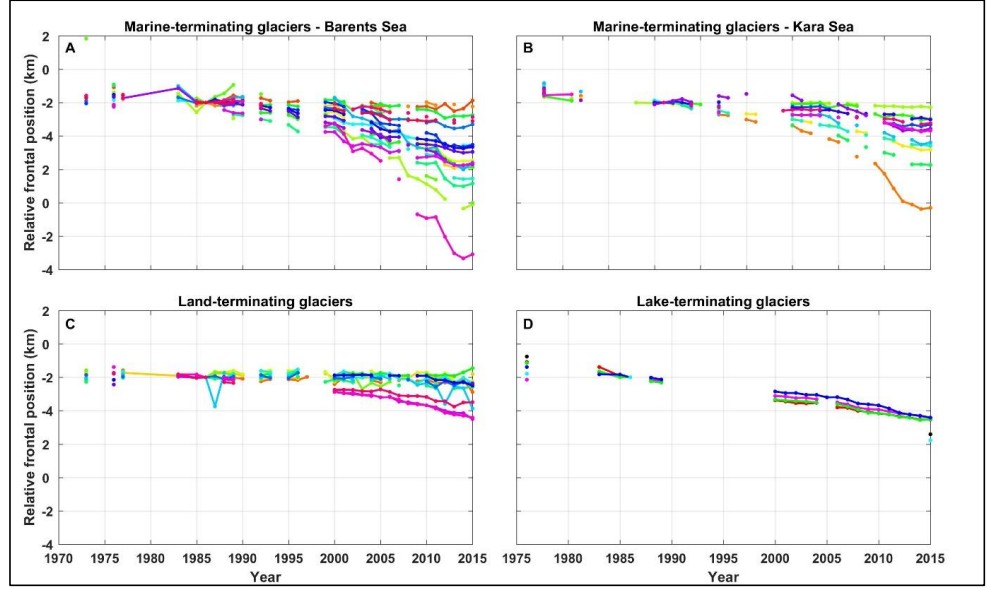

**Figure 5.** Relative glacier frontal position over time, from 1973 to 2015, for A) marine-terminating outlet glaciers on the Barents Sea coast; B) marine-terminating outlet glaciers on the Kara Sea coast; C) land-terminating outlet glaciers and D) Land-terminating outlet glaciers. Within each plot, frontal positions for each glacier are distinguished by different colours.

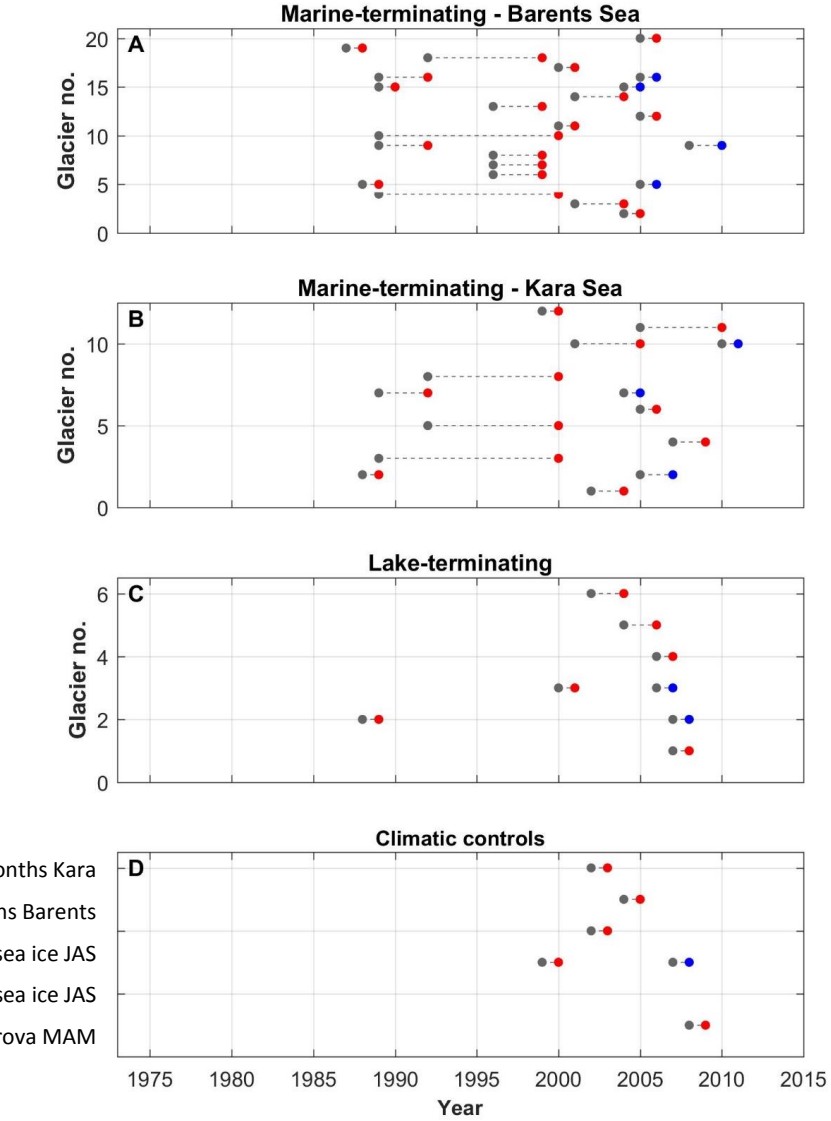

**Figure 6.** Results of the changepoint analysis for glacier retreat rates and climatic controls. Red dots indicate the start of a significantly different period in the time series data and grey dots represent the end of the previous period, with grey dashed lines connecting the two. This is done to account for missing data: we know that the changepoint occurred between the grey and the red dot, and that the new phase of behaviour occurred from the red dot onwards, but not the exact timing of the change. Blue dots show the start of a second significant change in the time series. Frontal position data were analysed separately for marine-terminating outlets on the Barents Sea (A), Kara Sea (B) coasts and lake-terminating glaciers (C). D) Changepoint results for seasonal means in air temperatures and sea ice, and the number of ice free months. Only climatic variables that demonstrated changepoints are shown.




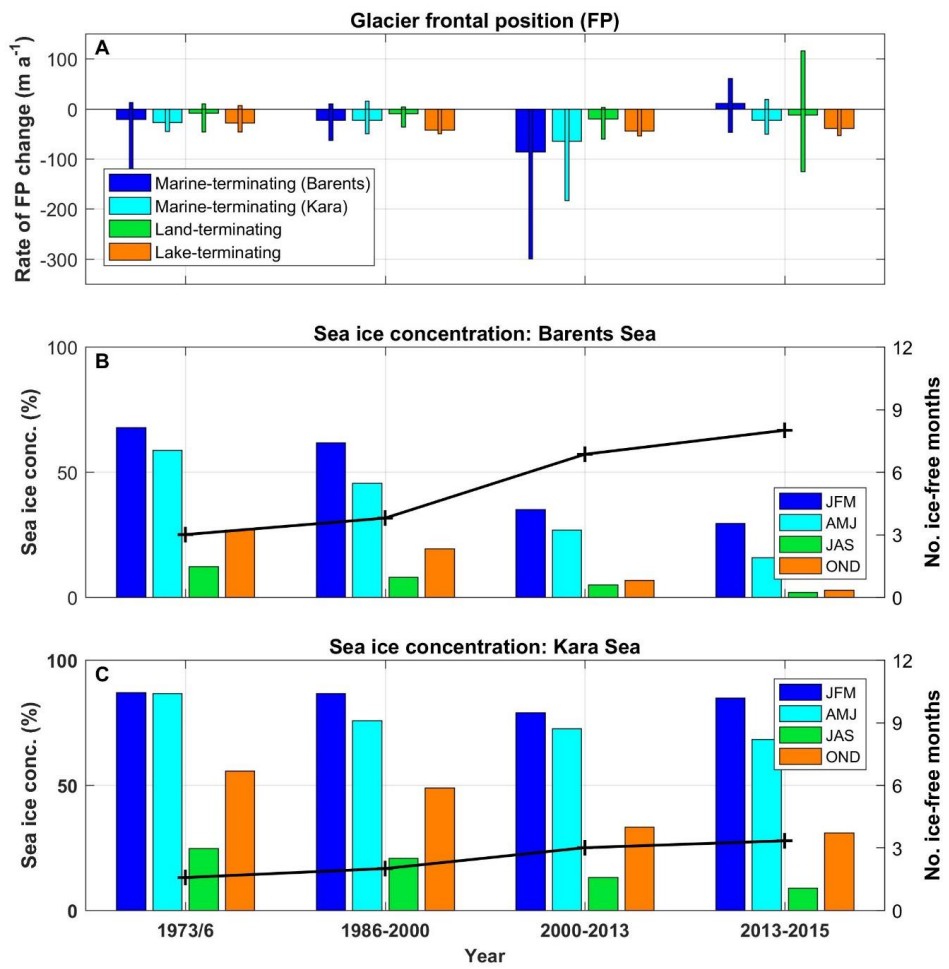

1018

**Figure 7.** Mean retreat rates for Novaya Zemlya outlet glaciers, and seasonal mean sea ice concentrations and
number of ice free months, for the Barents and Kara Sea coasts. Data are split into four time periods, based on
manually identified breaks in the glacier retreat data: 1973/6-1986, 1986-2000, 2000-2013 and 2013-2015. A)
Same as Fig. 4A. B & C) Mean seasonal sea ice concentrations (Jan-Mar, Apr-Jun, Jul-Sep and Oct-Dec) and
number of ice free months for the Barents Sea (B) and Kara Sea (C) coasts.

1024



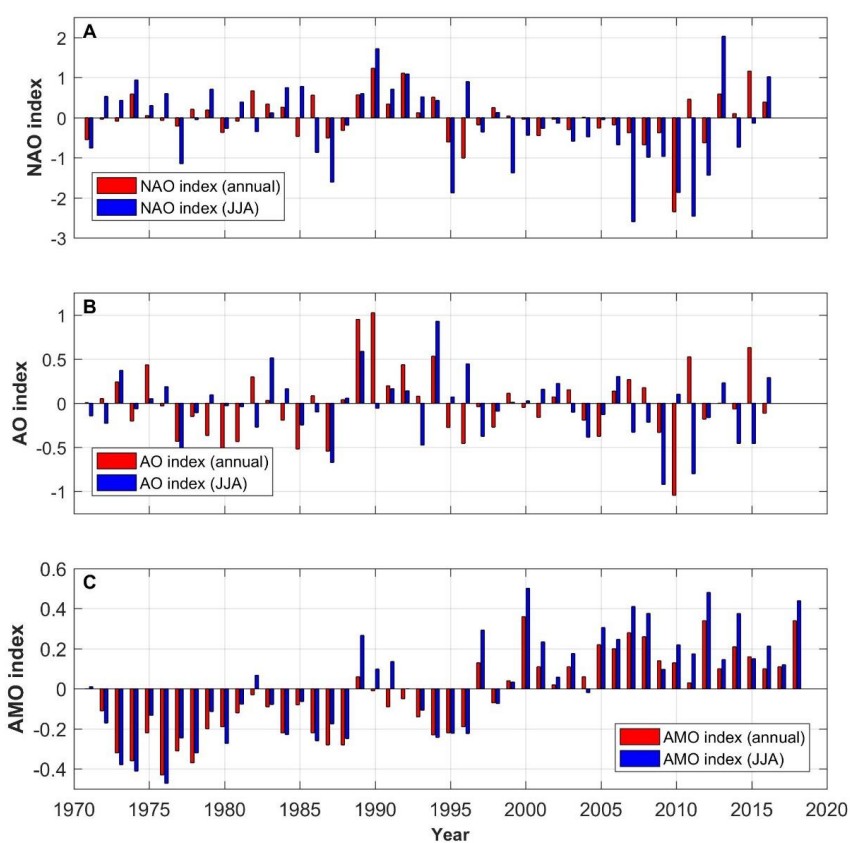

1025

**Figure 8.** Time series of A) North Atlantic Oscillation (NAO); B) Arctic Oscillation (AO); and C) Atlantic
Multidecadal Oscillation (AMO) for 1970 to 2016. In each case, mean annual and mean summer values are
shown.

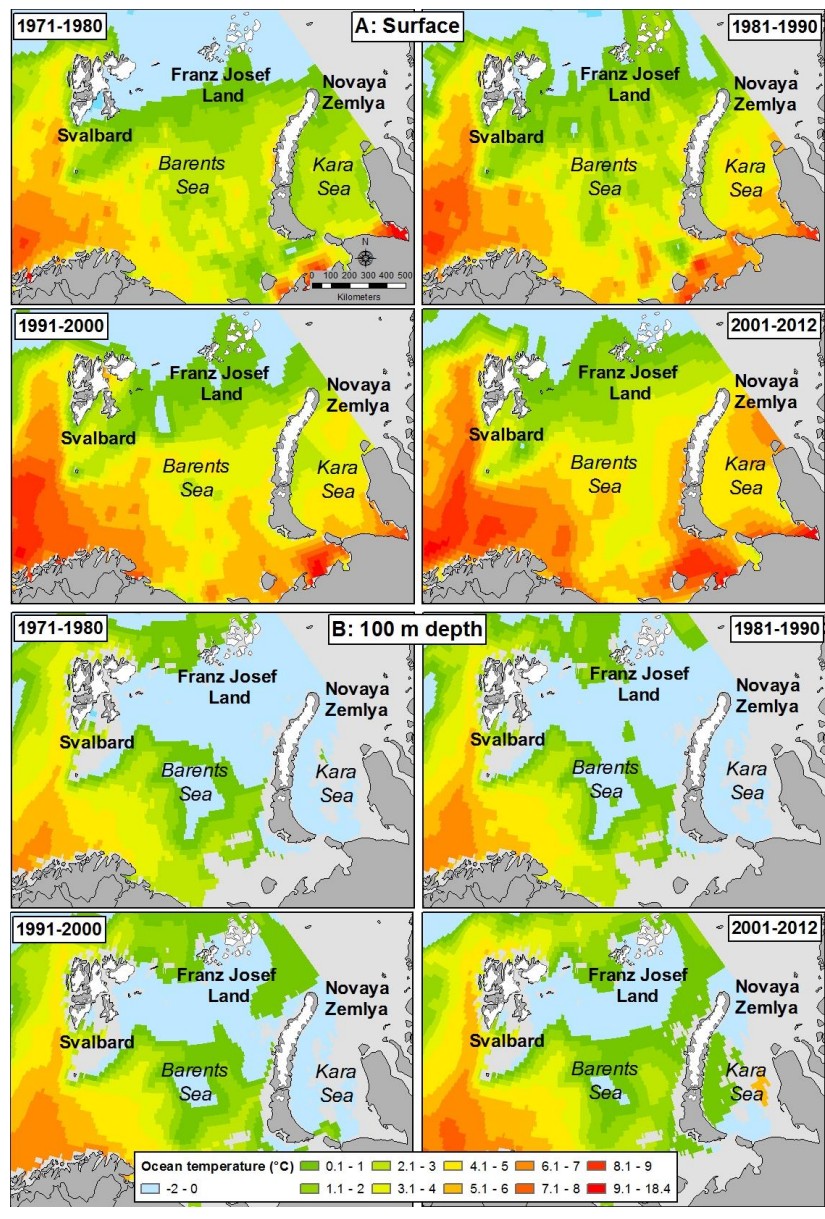

1029

**Figure 9.** Ocean temperatures from the 'Climatological Atlas of the Nordic Seas and Northern North Atlantic' (Korablev et al., 2014), at A) the surface and B) 100 m depth, for the following time intervals: 1971-1981, 1981-1991, 1991-2000 and 2001-2012. These intervals were chosen, to match as closely as possible with the glacier frontal position data and other datasets. Note that data coverage was substantially lower for 2001-2012, than compared to other time periods. Further details on data coverage are available here: https://www.nodc.noaa.gov/OC5/nordic-seas/.

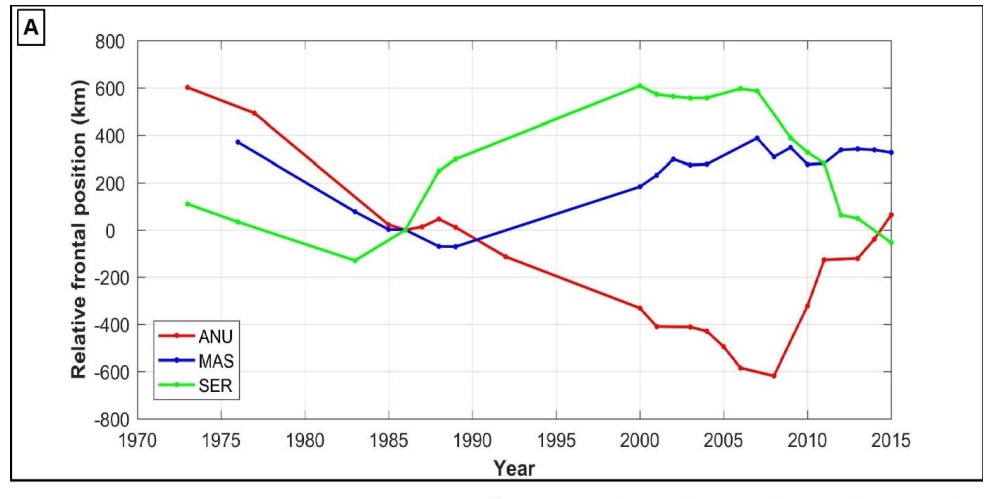

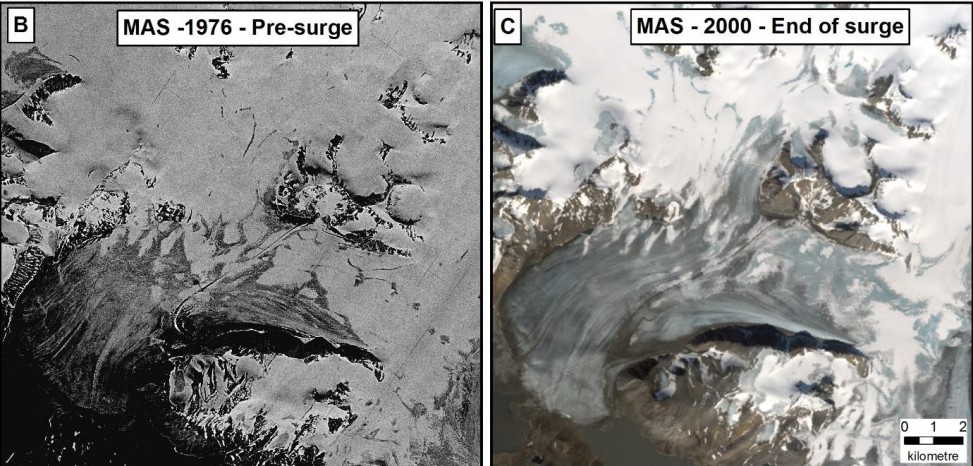

**Figure 10.** Glaciers identified as surging during the study period, based on the surge criteria compiled by Grant et al. (2009). A) Glacier frontal position (relative to 1986) for glaciers identified as surge type: Anuchina (ANU), Mashigina (MAS), and Serp i Molot (SER). B) Pre-surge imagery of MAS. Imagery source: Hexagon, 22nd July 1976. C) Imagery of MAS at the end of the surge. Imagery source: Landsat 7, 13th August 2000.