# Peer review of "Sea Ice - Barents"

_The Cryosphere, 2017_

## Referee Comment (RC1) · Anonymous Referee #1 · 15 Jun 2017

General comments. The paper describes marine-terminating glacier retreat on Novaya Zemlya (NVZ) between 1973/6 and 2015. That is, the content of the paper is much wider than its title, which rather reflects its main conclusion. This conclusion states (lines 680-682) that: "Retreat rates on marine-terminating glaciers were exceptional between 2000 and 2013, compared to previous decades. However, retreat slowed on the vast majority of ocean-terminating glaciers from 2013 onwards, and several glaciers advanced, particularly on the Barents Sea coast." In this regard my general questions are: (1) What are the intra-annual variations of glacier retreat rates on NVZ? (2) Are they comparable with the scale of deceleration observed in 2013-2015? (3) What are the trends and pattern in the NVZ glacier recession between 1973/6 and 2015

if evaluated not in linear measures but in area changes? It is highly desirable that data on the annual position of NVZ glacier fronts (presented now only in an unidentifiable form as different-color lines on Figure 5) will be available to readers as a separate tabular supplementary to the paper. The same is true for area changes if available.

Specific comments. line 57-58: Statement that "...the pattern of frontal position changes on NVZ prior to 1992 is uncertain and previous results indicate different trends..." seems to be to strong one, as all previous results indicate recession (Shumsky 1946, Chizov et al 1968, Koryakin 2013). line 90: It is not clear - does SER glacier belong to Sub 1 or to Northern ice mass? line 90: Total number of glaciers should be checked as data in the Table 1 (above the line 949) shows different number(s) - by terminus type: 32+6+15 = 53 and by ice mass: 43+4+5 = 52. line 118: "...The northern island also has two smaller ice caps, Sub 1 and Sub 2..." - There are not real ice caps but better say ice fields (or compound glacier systems). line 139: "Due to the lack of Landsat imagery during the 1990s..." contradicts with line 130 which states that data were available annually ...between 1985 and 1998. line 163: E.K. Fedorova but not E.K. Fedrova. Im. is an abbreviation from Russian word "imeni" which means "named after". To avoid ambiguity it seems better to indicate (here and everywhere in the text) the weather stations by WMO ID (20744 and 20946), as another weather station also named after E.K.Fedorov (WMO ID 20292) is located in other arctic place - on Cape Chelyuskin. line 169: Please, specify the data gaps on the Station Fedorova RSM00020946 lines 315-318: As shown by (Koryakin 2013) for NVZ glaciers there is some relation of retreat with their altitude. Also considering only the linear change of glaciers does not give full picture of their fluctuations. Analysis of area change of glaciers might reveal different aspects in fluctuation pattern/behavior/environmental control. line591-592: Observed reduction in retreat rates might be result from increased ice velocities. line 963: Strictly speaking the Northern ice cap is located to the north from INO. According the Russian nomenclature the Northern ice cap indicated on map is the Ice cap of Northern Island. line 973: it is not clear does the length of box "necks" mean something or not? Also there is no box at

Fig 2B for Kara L. Is it right? line 1003: Figure 5 is very interesting and most important, but its informativity is severely affected, since it is impossible to correspond the lines of different colors with specific glaciers (their names, or some other indicators, for example, RGI ID). line 1018: Thick black line is not specified in the caption of Figure 7.

Technical corrections. line 163 (and everywhere through the text): "Fedrova" should be "Fedorova". line 172 (and everywhere through the text, tables, figure captions, including text in supplementary file and title label placed on Supplementary Figure 1 B): "850 m" should be "850 mb". line 374: "+0.8 °C" should be "+0.8°C" (no space required). line 381: "7 %" should be "7%" (no space required). line 437: "SRE" should be "SER". line 992: title label at fig. 4C "Air Temperature: "Malaya Karmakuly" should be "Air Temperature: Malye Karmakuly". line 1031: "1981" should be "1980" line 1032: "1991" should be "1990". line 1036: label at vertical axes Fig. 10A "Relative frontal position (km)" should be "Relative frontal position (m)". Koryakin, V.S.: Glaciers of the New Earth in the XX century and global warming // [Priroda] Nature, 1, 42-48, 2013 [in Russian]. Shumsky, P.L.: Modern glaciation of the Soviet ArctiÑĄ [Sovremennoe oledenenie Sovetskoy Arktiki] // Moscow - Leningrad: 1949, 262 pp. [in Russian]. Chizhov, O.P., Koryakin, V.S, Davidovich, N.V., Kanevsky, Z.M., Zinger, E.M., Bazheva, V.Ya., Bazhev, A.B., and Khmelevskoy, I.F.: Glaciation of the New Earth [Oledenenie Novoy Zemli] // Moscow: Nauka, 1968, 338 pp. [in Russian].
* * *

---

## Referee Comment (RC2) · R. McNabb (Referee) · 19 Jun 2017

**Summary**

The authors have presented a record of glacier front positions for glaciers on Novaya Zemlya for the period covering 1975 - 2015. They have compared these changes with changes in air temperature, sea ice concentration, and climatological oscillations, analyzing the results with robust statistical methods. They conclude based on these results that the period 2000-2013 was significantly different for the marine-terminating glaciers, while other terminus types do not show significant changes throughout the time period.

[Figure]

The methods are well-described, the results well-presented and discussed, and the conclusions appear to be robust. As such, I have only a few minor comments, and I recommend the paper for acceptance pending these few comments.

Specific

**line 15:** delete "the" before "1973/76"

**lines 120-122:** These sentences are a little confusing to me. Consider emphasizing that these three glaciers were previously unknown to surge, if that is the case.

**lines 131-132:** What about orthorectification? It should not be much of a problem for tidewater glaciers, but land-based glacier termini significantly above sea level could be misplaced if the images are not orthorectified.

**lines 179-181:** How good an approximation is this to conditions near the glaciers?

**line 309, elsewhere:** I think there should be commas between $R2$ and p values.

**line 316:** If RHO is an acronym, it should be defined. If it is the Greek letter rho, use $\rho$ instead.

**line 432:** 18 years is an incredibly long time for an active phase!

**line 503:** linear relationship with latitude

**line 643:** Check the names here. It looks like MAS advances for 18 years (cf. also l. 432), SER advances for 15 years, and ANU begins surging in 2008.

**line 651:** Specify that the three glaciers you reference here are MAS, SER, and ANU, and not Tunabreen, Basin 3, and Variegated Glacier.

**lines 659,663:** I think you mean Fig. 10, and not Fig. 9. The large sediment plume is rather hard to see in Fig. 10c - you might consider enhancing this somehow. You could also make these into a separate figure, and include other images, say from 1985 and 1995, if they are available.

**Figure 5:** Fix the y-axis tick labels, as they should not go from 2 to -4 to 2 to -4 km.

**Figure 10a:** Relative frontal position in m, not km.

---

## Author Response (AR1)

We thank Reviewer 1 for their constructive and positive comments, which we feel have improved the paper. We have addressed all of the comments and provide our responses below, along with a reiteration of the comments, for reference.

The paper describes marine-terminating glacier retreat on Novaya Zemlya (NVZ) between 1973/6 and 2015. That is, the content of the paper is much wider than its title, which rather reflects its main conclusion. This conclusion states (lines 680-682) that: "Retreat rates on marine-terminating glaciers were exceptional between 2000 and 2013, compared to previous decades. However, retreat slowed on the vast majority of ocean-terminating glaciers from 2013 onwards, and several glaciers advanced, particularly on the Barents Sea coast."

In this regard my general questions are: (1) What are the intra-annual variations of glacier retreat rates on NVZ? (2) Are they comparable with the scale of deceleration observed in 2013-2015?

**RESPONSE: Seasonal variations are small, generally under 100 m (Carr et al., 2014). Assuming the calving season is 4 -6 months long, this would result in ~15-25 m of frontal variation in a month, which is below the image resolution. All of our imagery for 2012 – 2015 (i.e. from the end of more rapid retreat and through the slow down) are within 1 month of each other, meaning that any changes related to seasonal variations and differences in image data will be below the image resolution and therefore would not affect the results. The deceleration in retreat in 2013-2015 ranges from 35 m -to >120m, which is greater than any seasonal effects we may have inadvertently included by having different image dates. Furthermore, we have similar (and in some cases larger) gaps between imagery during the rapid retreat (2000-2013), and do not see any re-advances or seasonal trends, only continued retreat. Finally, the slow-down / advance persists across many glaciers (with slightly different image dates) and over three years (2013-2015), making it unlikely that it simply results from capturing part of the seasonal calving cycle.**

(3) What are the trends and pattern in the NVZ glacier recession between 1973/6 and 2015 if evaluated not in linear measures but in area changes?

**RESPONSE: We are focusing on glacier recession, not area change, as stated in the paper, and this has been done in many previous publications on Novaya Zemlya and elsewhere in the Arctic (e.g. Carr et al., 2014; Howat et al., 2008; McKnabb and Hock, 2014; Moon and Joughin, 2008). It is not simply linear change, in that we use a series of different time intervals and also use the box method, to account for uneven recession of the terminus. Even if area change were included, we do not think it would show substantially different patterns, as the main area of change would be at the terminus (as it is at the lowest altitude and in contact with the ocean / lakes). The vast majority of each glacier catchment (by area) is bounded by slower moving ice, belonging to the other glaciers, and therefore is unlikely to change over time. Any such changes would be very difficult to detect, even with accurate DEMs and velocity data, and changes in these ice divides would be the subject for another paper. As well as the main area of continuous ice, the glaciers have narrow tongues, reaching down to the sea. Particularly on the Barents Sea coast, many of these are bounded by moraines / higher ground, meaning that any lateral changes would be limited. Where the glaciers are less constrained by topography, we would expect ice loss to reduce with elevation anyway, due to the altitudinal lapse rate, meaning that changes should be maximum at the termini. As stated above, we focus on terminus change in this paper, as previous studies have highlight its importance for driving dynamic changes, such as ice acceleration and dynamic thinning (e.g. Pritchard et al., 2009; Howat et al., 2007; Joughin et al., 2004), as well as its quick response to changes in forcing (e.g. Carr et al., 2013). In contrast, changes in area would reflect processes operating on a range of time scales, from rapid terminus response to e.g. ocean warming, through to longer-term surface mass balance change, and it would be difficult to separate these out. As such, glacier retreat, as opposed to area change, is the most appropriate measure for our study.**

It is highly desirable that data on the annual position of NVZ glacier fronts (presented now only in an unidentifiable form as different-color lines on Figure 5) will be available to readers as a separate tabular supplementary to the paper. The same is true for area changes if available. **RESPONSE: We have added these**

**data to the supplementary information (Supp. Tables 3-6), along with a table detailing the glacier ID, Randolph Glacier Inventory ID and name, where available (Supp. Table 1). Area changes are not available.**

Specific comments. line 57-58: Statement that ": : :the pattern of frontal position changes on NVZ prior to 1992 is uncertain and previous results indicate different trends: : :" seems to be to strong one, as all previous results indicate recession (Shumsky 1946, Chizov et al 1968, Koryakin 2013). **RESPONSE: As referenced in the text, Zeeberg and Forman (2001) showed that half the glaciers on north Novaya Zemlya were stable between 1964 and 1993, so not all previous studies indicate recession. We have added the papers referenced here.**

line 90: It is not clear - does SER glacier belong to Sub 1 or to Northern ice mass? **RESPONSE: It belongs to the northern ice mass. However, it does not matter for the paper, as it is surge type, so excluded from the assessment of glacier response to climate (Line 122).**

line 90: Total number of glaciers should be checked as data in the Table 1 (above the line 949) shows different number(s) - by terminus type: 32+6+15 = 53 and by ice mass: 43+4+5 = 52. **RESPONSE: Corrected. The numbers in the table were updated.**

line 118: ": : :The northern island also has two smaller ice caps, Sub 1 and Sub 2: : :" - There are not real ice caps but better say ice fields (or compound glacier systems). **RESPONSE: Agreed. We now use the term 'ice field' or 'ice mass' throughout.**

line 139: "Due to the lack of Landsat imagery during the 1990s: : :" contradicts with line 130 which states that data were available annually ...between 1985 and 1998. **RESPONSE: Line 130 should say 'between 1985 and 1989'. This has been corrected.**

line 163: E.K. Fedorova but not E.K. Fedrova. Im. is an abbreviation from Russian word "imeni" which means "named after". To avoid ambiguity it seems better to indicate (here and everywhere in the text) the weather stations by WMO ID (20744 and 20946), as another weather station also named after E.K.Fedorov (WMO ID 20292) is located in other arctic place - on Cape Chelyuskin. **RESPONSE: IM. Was removed. We have added the WMO ID's here as suggested, but continue to use the names throughout the text, as readers unfamiliar with the numbers may otherwise need to keep referring back. Adding the WMO IDs here removes the ambiguity about the other, similarly named station. WMO IDs have also been added to the captions for Fig. 1 & 4, and to Supp Table 1, for clarity.**

line 169: Please, specify the data gaps on the Station Fedorova RSM00020946. **RESPONSE: Seasonal averages were only calculated where data were available for all months and, by extension, annual averages were only calculated where all months of the year were available. This has been added (Line: 186). It would become very long-winded to specify every data gap in the text, so we have added the meteorological data as Supplementary Table 2, so that those who are interested can see the gaps.**

lines 315-318: As shown by (Koryakin 2013) for NVZ glaciers there is some relation of retreat with their altitude. Also considering only the linear change of glaciers does not give full picture of their fluctuations. Analysis of area change of glaciers might reveal different aspects in fluctuation pattern/behavior/environmental control. **RESPONSE: Here we focus on latitude and catchment area, as opposed to altitude, as we are looking at changes at the glacier termini. Most of the glaciers are marine-terminating, and therefore terminate at sea level, so this would not help us to assess controls on retreat patterns. We agree that looking at the overall change does not necessarily give a full picture of their fluctuations. However, this is assessed later in the paper, via the change point analysis and by presenting the time series for each glacier. The aim here was to see if latitude controlled overall retreat rate and our results show this was not the case. Similarly, our data show large variability in retreat rates at a range of time steps (e.g. Figs. 4 & 5), which also does not appear to relate to latitude. We do not think that looking at area would substantially effect the results, as outlined above.**

Line 591-592: Observed reduction in retreat rates might be result from increased ice velocities. **RESPONSE: This is a possibility. However, with available data it is not possible to determine whether increased ice velocities are a response to rapid retreat, or whether reduced retreat is due to more rapid delivery of ice**

to the calving front. In either case, our point here is that the changes relate to the dynamics of the outlet glaciers, rather than upstream changes in the surface mass balance. Data on surface elevation change and ice velocities are also needed to understand the short-term dynamic behaviour of these outlet glaciers. However, this goes beyond the scope of this paper, and would be another paper in itself. We have added a sentence to this effect at Line 621.

line 963: Strictly speaking the Northern ice cap is located to the north from INO. According the Russian nomenclature the Northern ice cap indicated on map is the Ice cap of Northern Island. **RESPONSE: Thank you, we did not know this. In the text, we have stated that the name is 'ice cap of the northern island' (Line 89), but that we refer to it as the 'northern island ice cap' for brevity. We have updated the maps and figures accordingly.**

line 973: it is not clear does the length of box "necks" mean something or not? Also there is no box at Fig 2B for Kara L. Is it right? **RESPONSE: We are not entirely sure what is meant here, but as stated in the caption, the red line is the mean and the blue lines are the upper and lower quartiles, meaning that the length between the two blues lines is the inter-quartile range. If the reviewer is referring to the differences in the width of the red line between the different sub-plots, this is simply because there are four categories in B, compared to three categories in A & C, so the bars need to be narrower to fit on the plot. For Kara L, this was incorrect and due to some trailing zeros in the data. It has been corrected. Thanks for highlighting this.**

line 1003: Figure 5 is very interesting and most important, but its informativity is severely affected, since it is impossible to correspond the lines of different colors with specific glaciers (their names, or some other indicators, for example, RGI ID). **RESPONSE: See above.**

line 1018: Thick black line is not specified in the caption of Figure 7. **RESPONSE: Corrected**

**Technical corrections.**

line 163 (and everywhere through the text): "Fedrova" should be "Fedorova". **RESPONSE: Corrected**

line 172 (and everywhere through the text, tables, figure captions, including text in supplementary file and title label placed on Supplementary Figure 1 B): "850 m" should be "850 mb". **RESPONSE: The units should be hPa and this has been corrected throughout.**

line 374: "+0.8 ∘C" should be "+0.8∘C" (no space required). **RESPONSE: No, following conventions for SI units, there should be a space between the numeric value and the unit. E.g. See http://ukma.org.uk/docs/ukma-style-guide.pdf.**

line 381: "7 %" should be "7%" (no space required). **RESPONSE: See above.**

line 437: "SRE" should be "SER". **RESPONSE: Corrected**

line 992: title label at fig. 4C "Air Temperature: "Malaya Karmakuly" should be "Air Temperature: Malye Karmakuly". **RESPONSE: Corrected**

line 1031: "1981" should be "1980" **RESPONSE: Corrected**

line 1032: "1991" should be "1990". **RESPONSE: Corrected**

line 1036: label at vertical axes Fig. 10A "Relative frontal position (km)" should be "Relative frontal position (m)". **RESPONSE: Corrected**

We thank Robert McNabb for his constructive and very positive comments on the paper. We have addressed all of the comments and provide our responses below, along with a reiteration of the comments, for reference.

**Summary**

The authors have presented a record of glacier front positions for glaciers on Novaya Zemlya for the period covering 1975 - 2015. They have compared these changes with changes in air temperature, sea ice concentration, and climatological oscillations, analyzing the results with robust statistical methods. They conclude based on these results that the period 2000-2013 was significantly different for the marine-terminating glaciers, while other terminus types do not show significant changes throughout the time period.  The methods are well-described, the results well-presented and discussed, and the conclusions appear to be robust. As such, I have only a few minor comments, and I recommend the paper for acceptance pending these few comments.

**RESPONSE: We thank you very much for your positive comments regarding the paper and for the minor improvements suggested below.**

**Specific**

**line 15:** delete "the" before "1973/76"

**RESPONSE: Updated.**

**lines 120-122:** These sentences are a little confusing to me. Consider emphasizing that these three glaciers were previously unknown to surge, if that is the case.

**RESPONSE: Two of the glaciers were known to surge, but our data better constrains the timing, and the third was suggested to surge and we show it surging for the first time. We have revised the text to clarify (Lines 122-130).**

**lines 131-132:** What about orthorectification? It should not be much of a problem for tidewater glaciers, but land-based glacier termini significantly above sea level could be misplaced if the images are not orthorectified.

**RESPONSE: We do not believe that orthorectification is required here. The terrain is relatively gentle and not mountainous around these termini, unlike areas such as the Himalaya or the Alps, where glaciers are constrained in high-sided valleys. As such, orthorectification is unlikely to make any discernible difference. We also checked each of the manually georeferenced images against Landsat 8 imagery (which we took as the most likely to be accurately georeferenced), to ensure that they matched correctly, for both land- and marine-terminating glaciers. We did this by matching up features that should not move (e.g. large rock fractures) close the glacier termini and also checking for any unexpectedly large changes in the glacier margins. We rejected any images where we saw movement of features that should be static and/or where the glaciers were clearly incorrectly located. As such, we are confident that the geoferencing was sufficient for the marine- and land-terminating glaciers here and that the images are co-located as closely as the imagery resolution allows. We have added a brief explanation of this at Lines 143-150.**

**Lines 179-181:** How good an approximation is this to conditions near the glaciers?

**RESPONSE: This is the best approximation we have. We wanted to use the same dataset for the entire time series, to ensure consistency, which means we had to compromise on the spatial resolution. NVZ glaciers are relatively exposed to the open ocean and do not have long winding fjords. As such, conditions immediately offshore are likely to be reasonably representative. In an ideal world, we would have data directly from the glacier front, but it is not possible over these time scales. We have added text to this effect (Line 198-203)**

**Line 309, elsewhere:** I think there should be commas between R2 and p values.

**RESPONSE: Yes, agreed. We have updates this throughout.**

**Line 316:** If RHO is an acronym, it should be defined. If it is the Greek letter rho, use $\rho$ instead.

**RESPONSE: Yes, agreed. It should be the Greek letter rho.**

**line 432:** 18 years is an incredibly long time for an active phase!

**RESPONSE: Yes, we agree. This was one of the justifications for including the surge-type glaciers in the paper, as it seemed incredibly long. It may be even longer, as we are only looking at terminus change here. We suspect it may be towards the end member of surging, possibly due to low mass turnover, comparatively cold conditions and the glaciers being polythermal. We do not know about the substrate, but this may also contribute. We wanted to note these characteristics and believe it would be an interesting focus for follow up work.**

**line 503:** linear relationship with latitude

**RESPONSE: Updated.**

**line 643:** Check the names here. It looks like MAS advances for 18 years (cf. also l. 432), SER advances for 15 years, and ANU begins surging in 2008.

**RESPONSE: Updated.**

**line 651:** Specify that the three glaciers you reference here are MAS, SER, and ANU, and not Tunabreen, Basin 3, and Variegated Glacier.

**RESPONSE: Updated.**

**lines 659,663:** I think you mean Fig. 10, and not Fig. 9. The large sediment plume is rather hard to see in Fig. 10c - you might consider enhancing this somehow. You could also make these into a separate figure, and include other images, say from 1985 and 1995, if they are available.

**RESPONSE: Figure numbers have been updated. As suggested, we have added in imagery from other time points, to show the surge progression in more detail. Specifically, we show pre-surge (1976), surge of the tributary (1985-1988) and surge of the main front (2000). We show the maximum terminus extent in 2007. The image dates are the best available. We have also added a sub—figure, showing the plumes from ANU, which are more obvious than those from MAS.**

**Figure 5:** Fix the y-axis tick labels, as they should not go from 2 to -4 to 2 to -4 km.

**RESPONSE: Updated.**

**Figure 10a:** Relative frontal position in m, not km.
**RESPONSE: Updated.**

[revised manuscript text omitted]

island (hereafter referred to as the northern island ice cap for brevity) and its subsidiary ice  fields (Fig. 1).
We were unable to find the names of these subsidiary ice  fields in the literature, so we name them Sub 1
and Sub 2 (Fig. 1). A total of 54 outlet glaciers were investigated, which allowed us to assess the impact of
different glaciological, climatic and oceanic settings on retreat (Supp. Table 1). Specifically, we assessed the
impact of coast (Barents versus Kara Sea on the northern ice mass), ice mass ( northern island ice cap,
Sub 1 or Sub 2), terminus type (marine-, lake- and land-terminating) and latitude (Table 1). The two coasts of
Novaya Zemlya are characterised by very different climatic and oceanic conditions: the Barents Sea coast is
influenced by water from the north Atlantic (Loeng, 1991; Pfirman et al., 1994; Politova et al., 2012) and subject
to Atlantic cyclonic systems (Zeeberg and Forman, 2001), which results in warmer air and ocean temperatures as
well as higher precipitation (Przybylak and Wyszyński, 2016; Zeeberg and Forman, 2001). In contrast, the Kara
Sea coast is isolated from north Atlantic weather systems, by the topographic barrier of NVZ (Pavlov and Pfirman,
1995), and is subject to cold, Arctic-derived water, along with much higher sea ice concentrations (Zeeberg and
Forman, 2001). We therefore aim to investigate whether these differing climatic and oceanic conditions lead to
major differences in glacier retreat between the two coasts. Glaciers identified as surge-type (Grant et al., 2009)
were excluded from the retreat calculations and analysis. However, frontal position data are presented separately
for three glaciers that were actively surging during the study period. Glacier retreat was assessed from the 1973/6
to 2015, in order to provide the greatest temporal coverage possible from satellite imagery. We use these data to
address the following questions:

1. At multi-decadal timescales, is there a significant difference in glacier retreat rates according to: i)
   terminus type (land-, lake- or marine-terminating); ii) coast (Barents versus Kara Sea coast); iii) ice mass
   (northern ice mass, Sub 1 or Sub 2) and; iv) latitude?

2. Are outlet glacier retreat rates observed between 2000 and 2010 on NVZ exceptional during the past ~
years?

3. Is glacier retreat accelerating, decelerating or persisting at the same rate?

4. Can we link observed retreat to changes in external forcing (air temperatures, sea ice and/or ocean
temperatures)?

**2. Methods**

**2.1. Study area**

This paper focuses on the ice masses located on the Severny Island, which is the northern island of the Novaya
Zemlya archipelago (Fig. 1). The northern island ice cap contains the vast majority of ice (19,841 km$^2$) and the
majority of the main outlet glaciers (Fig. 1). The northern island also has two smaller ice fields, Sub 1 and
Sub 2, which are much smaller in area (1010 km$^2$ and 705 km$^2$ respectively) and have far fewer, smaller outlet
glaciers (Sub 1 = 4; Sub 2 = 5) (Fig. 1). All glaciers that have been previously identified as surge
type and those smaller than 1 km in width were excluded from our main analysis of glacier retreat rates and
response to climate forcing. However, we also observed three glaciers surging during the study period: ANU,
MAS and SER (Fig. 1). MAS and SER have been previously identified as surge type (Grant et al., 2009), but our
data provides better constraints on the duration and timing of these surges. ANU was identified as potentially
surge-type, on the basis of looped moraines (Grant et al., 2009). Our study confirms it as surge-type and provides
information on the surge timing and duration. These three glaciers are not included in the assessment of NVZ
glacier response to climate change, as surging can occur impudently of climate forcing (Meier and Post, 1969),
but are discussed separately, to improve our knowledge of NVZ surge characteristics.
This resulted in a total of 54 outlet glaciers, which were
located in a variety of settings and hence allowed us to assess spatial controls on glacier retreat (Table 1). Where
available glacier names and World Glacier Inventory IDs are given in Supplementary Table 1, along with glacier
acronyms used in this paper. The impact of coast could only be assessed for the main ice mass, as the glaciers on
the smaller ice masses, Sub 1 and Sub 2, are located on the southern ice margin so do not fall on either coast (Fig.
1).

**2.2. Glacier frontal position**
Outlet glacier frontal positions were acquired predominantly from Landsat imagery. These data have a spatial
resolution of 30 m and were obtained freely via the United States Geological Survey (USGS) Global Visualization
Viewer (Glovis) (http://glovis.usgs.gov/). The frequency of available imagery varied considerably during the
study period. Data were available annually from 1999 to 2015 and between 1985 and 19,89 although
georeferencing issues during the latter time period meant that imagery needed to be re-coregistered manually
using stable, off-ice locations as tie-points. Prior to 1985, the only available Landsat scenes dated from 1973, and
these also needed to be manually georeferenced. We verified all images that required georeferencing against
Landsat 8 data, which should have the most accurate location data of the imagery timeseries. We did this by
comparing the location of features that should be static between images (e.g. large rock fractures) and also
checking for any unrealistic changes in the lateral glacier margins, over and above what could be expected by
glacier melting. Any images where we saw changes in the location of static features, above the image resolution
were not used. As such, orthorectification was not required for these images, as the terrain is relatively gentle on
NVZ and our verification process showed that the images were co-located with the Landsat 8 imagery to within
a pixel using just georeferencing. Hexagon KH-9 imagery was used to determine frontal positions in 1976 and

1977, but full coverage of the study area was not available for either year. The data resolution is 20 to 30 feet (~6-9 m). The earliest common date for which we have frontal positions for all glaciers is 1986, and so we calculate total retreat rates for the period 1986-2015 and use these values to assess spatial variability in glacier recession across the study region. All glacier frontal positions are calculated relative to 1986 (i.e. the frontal position in 1986 = 0 m), to allow for direct comparison.

[revised manuscript text omitted]

Sea ice data were acquired from the Nimbus-7 SMMR and DMSP SSM/I-SSMIS Passive Microwave dataset (https://nsidc.org/data/docs/daac/nsidc0051_gsfc_seaice.gd.html). The data provide information on the percentage of the ocean covered by sea ice and this is measured using brightness temperatures from microwave sensors. The data have a spatial resolution of 25 x 25 km and we use the monthly-averaged product. This dataset was selected due to its long temporal coverage, which extends from 26 October 1978 to 31 December 2015 and thus provides a consistent dataset throughout our study period. NVZ glaciers are not located within long fjords and are relatively exposed to the open ocean (Fig. 1). Consequently, sea ice conditions within 25 km of the glacier fronts (i.e. the data resolution) are likely to be reasonably representative of the overall sea ice trends experienced by the glaciers, particularly at the decadal time scales assessed here. However, it should be noted that the data cannot provide detailed information on sea ice conditions specific to each glacier front, but are used here, as they are the only dataset available for the entire 
[revised manuscript text omitted]
 & F-C). The exact timing of this tributary surge is uncertain, but imagery from 1985 (Fig. 10C) shows
limited evidence of surging, whereas a number of surge indicators are clearly visible by 1988, including looped
moraines and rapid advance (Fig. 10D), suggesting it began in the late 1980s. The tributary glacier This ice appears
then to have impacted onadvanced into the eastern margin of the main outlet of MAS, causing glacier it to advance,
and produced heavy crevassing on the eastern portion of its terminus (Figs. 10BD & CE). The main terminus of
MAS reached its maximum extent for the study period in 2007, and the tributary continued advancing from the
1980s until 2007 (Fig. 10 F). The role of the tributary glacier in triggering the surge 
[revised manuscript text omitted]

Shumsky, P. L.: Modern glaciation of the Soviet Arctic, [Sovremennoe oledenenie Sovetskoy Arktiki]
Moscow - Leningrad [in Russian], 1949.
Sole, A., Payne, T., Bamber, J., Nienow, P., and Krabill, W.: Testing hypotheses of the cause of
peripheral thinning of the Greenland Ice Sheet: is land-terminating ice thinning at anomalously high
rates?, The Cryosphere Discussions, 2, 673–710, 2008.

Sutherland, D. A., Straneo, F., Stenson, G. B., Davidson, F., Hammill, M. O., and Rosing-Asvid, A.:
Atlantic water variability on the SE Greenland shelf and its relationship to SST and bathymetry,
Journal of Geophysical Research-Oceans, doi: 10.1029/2012JC008354, 2013. 2013.
Sutton, R. T. and Hodson, D. L.: Atlantic Ocean forcing of North American and European summer
climate, science, 309, 115-118, 2005.
Trüssel, B. L., Motyka, R. J., Truffer, M., and Larsen, C. F.: Rapid thinning of lake-calving Yakutat
Glacier and the collapse of the Yakutat Icefield, southeast Alaska, USA, Journal of Glaciology, 59,
149-161, 2013.
van den Broeke, M., Bamber, J., Ettema, J., Rignot, E., Schrama, E., van de Berg, W. J., van Meijgaard,
E., Velicogna, I., and Wouters, B.: Partitioning Recent Greenland Mass Loss, Science, 326, 984-986,
2009.
van der Veen, C. J.: Fracture mechanics approach to penetration of bottom crevasses on glaciers,
Cold Regions Science and Technology, 27, 213- 223, 1998a.
van der Veen, C. J.: Fracture mechanics approach to penetration of surface crevasses on glaciers Cold
Regions Science and Technology, 27, 31-47, 1998b.
Willis, M. J., Melkonian, A. K., and Pritchard, M. E.: Outlet glacier response to the 2012 collapse of
the Matusevich Ice Shelf, Severnaya Zemlya, Russian Arctic, Journal of Geophysical Research: Earth
Surface, 120, 2040-2055, 2015.
Zeeberg, J. and Forman, S. L.: Changes in glacier extent on north Novaya Zemlya in the Twentieth
Century, The Holocene, 11, 161-175, 2001.
Zhao, M., Ramage, J., Semmens, K., and Obleitner, F.: Recent ice cap snowmelt in Russian High Arctic
and anti-correlation with late summer sea ice extent, Environmental Research Letters, 9, 045009,
2014.
Zhou, S., Miller, A., JWang, J., and Angell, J.: Trends of NAO and AO and their associations with
stratospheric processes, Geophysical Research Letters, 28, 4107-4110, 2001.

Formatted Table

| Characteristic | Category | Number of glaciers |
|---|---|---|
| **Coast** | Barents Sea | 27 |
| | Kara Sea | 18 |
| **Ice mass** | Northern island ice cap | 43 |
| | Subsidiary ice mass 1 | 4 |
| | Subsidiary ice mass 2 | 5 |
| **Terminus type** | Marine | 34 |
| | Lake | 6 |
| | Land | 1 |

**Table 1**. Number of outlet glaciers contained within each category used to assess spatial variations in retreat rate, specifically coast, ice mass and terminus type.

| | Barents Sea marine-terminating | Kara Sea marine-terminating | Land-terminating | Lake-terminating |
|---|---|---|---|---|
| 76-86 / 86-00 | 0.440 | 0.538 | 0.982 | 0.486 |
| 76-86 / 00-13 | **>0.001** | **0.018** | 0.085 | 0.686 |
| 76-86 / 13-15 | **0.008** | 0.497 | 0.945 | 0.686 |
| 86-00 / 00-13 | **0.001** | **0.008** | 0.223 | 0.886 |
| 86-00 / 13-15 | **0.001** | 0.935 | 0.909 | 0.886 |
| 00-13 / 13-15 | **>0.001** | **0.009** | 0.597 | 0.686 |

**Table 2.** Wilcoxon test results, used to assess significant differences in retreat rates between each manually- identified time interval (1976-1986, 1986-2000, 2000-2013, 2013, 2015). Retreat rate data were tested separately for each terminus type, and marine-terminating glaciers were further sub-divided by coast. Following convention, p-values of <0.05 are considered significant and are highlighted in bold.

| Station | Time interval | Season | | | | |
|---|---|---|---|---|---|---|
| | | DJF | MAM | JJA | SON | Annual |
|  E.K. Fedorova | 13-15 / 86-99 | 0.432 | | | | |
|  E.K. Fedorova | 13-15 / 76-85 | 0.937 | | | | |
|  E.K. Fedorova | 00-12 / 13-15 | 0.287 | | | | |
|  E.K. Fedorova | 00-12 / 86-99 | **0.011** | 0.643 | **0.043** | **0.008** | **0.013** |
|  E.K. Fedorova | 00-12 / 76-85 | 0.186 | **0.035** | **0.045** | **0.003** | **0.003** |
|  E.K. Fedorova | 86-99 / 76-85 | 0.188 | 0.089 | 0.704 | 0.495 | 0.828 |
| | | | | | | |
| Malye Karmakuly | 13-15 / 86-99 | | | | | |
| Malye Karmakuly | 13-15 / 76-85 | | | | | |
| Malye Karmakuly | 00-12 / 13-15 | | - | - | - | - |

| Malye Karmakuly | 00-12 / 86-99 | 0.017 | 0.840 | 0.056 | **0.007** | **0.017** |
| Malye Karmakuly | 00-12 / 76-85 | **0.038** | **0.041** | **0.045** | **0.004** | **>0.001** |
| Malye Karmakuly | 86-99 / 76-85 | 0.623 | 0.086 | 0.5977 | 0.673 | 0.212 |
| | | | | | | |
| ERA-Interim (surface) | 13-15 / 86-99 | **0.032** | 0.156 | 0.197 | 0.156 | **0.006** |
| ERA-Interim (surface) | 13-15 / 76-85 | 0.714 | 0.083 | 0.517 | 0.833 | 0.117 |
| ERA-Interim (surface) | 00-12 / 13-15 | 0.900 | 0.189 | 0.364 | 0.593 | 0.239 |
| ERA-Interim (surface) | 00-12 / 86-99 | **0.006** | 0.942 | 0.981 | 0.062 | **0.044** |
| ERA-Interim (surface) | 00-12 / 76-85 | 0.765 | 0.579 | 0.526 | 0.874 | 0.267 |
| ERA-Interim (surface) | 86-99 / 76-85 | 0.127 | 0.233 | 0.970 | 0.192 | 0.794 |
| | | | | | | |
| ERA-Interim (850 hPa) | 13-15 / 86-99 | 0.591 | 0.509 | 0.432 | 0.500 | 0.206 |
| ERA-Interim (850 hPa) | 13-15 / 76-85 | 0.548 | 0.383 | 0.833 | 0.733 | 0.383 |
| ERA-Interim (850 hPa) | 00-12 / 13-15 | 0.521 | 0.611 | 0.782 | 0.511 | 0.900 |
| ERA-Interim (850 hPa) | 00-12 / 86-99 | 0.062 | 0.752 | 0.058 | **0.041** | **0.004** |
| ERA-Interim (850 hPa) | 00-12 / 76-85 | 0.831 | 0.303 | 0.939 | 0.751 | 0.132 |
| ERA-Interim (850 hPa) | 86-99 / 76-85 | 0.149 | 0.433 | 0.433 | 0.146 | 0.576 |
| | | | | | | |

[revised manuscript text omitted]
). Tributary prior to the appearance of obvious surge-type features. Imagery source: Landsat 5, 26th July 1985. CD) Imagery of MAS at the end of the surgeduring the surge of its tributary. Imagery source: Landsat 57, 13th August13th August 20001988. E) MAS during the surge of the main glacier trunk. Imagery source: Landsat 7, 13th August 2000. F) MAS at the end of main glacier the surge, showing the maximum observed extent of the main terminus. Imagery source: Landsat 7, 8th July 2007. G) Sediment plumes emerging from the margin of ANU during its recent surge. Imagery source: Landsat 8, 31st July 2015.